# Wavelet-Driven Masked Multiscale Reconstruction for PPG Foundation Models

## Abstract

Wearable foundation models have the potential to transform digital health by learning transferable representations from large-scale biosignals collected in everyday settings. While recent progress has been made in large-scale pretraining, most approaches overlook the spectral structure of photoplethysmography (PPG) signals, wherein physiological rhythms unfold across multiple frequency bands. Motivated by the insight that many downstream health-related tasks depend on multi-resolution features spanning fine-grained waveform morphology to global rhythmic dynamics, we introduce Masked Multiscale Reconstruction (MMR) for PPG representation learning – a self-supervised pretraining framework that explicitly learns from hierarchical time–frequency scales of PPG data. The pretraining task is designed to reconstruct randomly masked out coefficients obtained from a wavelet-based multiresolution decomposition of PPG signals, forcing the transformer encoder to integrate information across temporal and spectral scales. We pretrain our model with MMR using ∼17 million unlabeled 10-second PPG segments from ∼32,000 smartwatch users. On 17 of 19 diverse health-related tasks, MMR trained on large-scale wearable PPG data outperforms or matches state-of-the-art open-source PPG foundation models, time-series foundation models and other self-supervised baselines. Extensive analysis of our learned embeddings and systematic ablations underscore the value of wavelet-based representations, showing that they capture robust and physiologically-grounded features. Together, these results highlight the potential of MMR as a step toward generalizable PPG foundation models.

## 1 Introduction

Foundation models for biosignals remain in their infancy, despite early promise demonstrating their potential to transform health monitoring and biomarker discovery (Abbaspourazad et al., 2024b; Narayanswamy et al., 2024; Erturk et al., 2025; Xu et al., 2025). Among these signals, photoplethysmography (PPG) is uniquely well-suited for self-supervised learning: it is embedded in virtually every consumer wearable, already underpins multiple deployed machine learning models for applications such as blood pressure, arrhythmia, and stress detection (Song et al., 2019; Bashar et al., 2019; Namvari et al., 2022; Apple Inc., 2025), and offers large-scale data for continuous cardiovascular monitoring (Charlton et al., 2022a; Lee & Akamatsu, 2025). Recent PPG foundation models (Abbaspourazad et al., 2024a; Pillai et al., 2024; Saha et al., 2025) have demonstrated that self-supervised pre-training can outperform traditional ML approaches, underscoring the promise of large-scale pretraining paradigms. More broadly, recent time-series self-supervised models have shown that explicitly modeling spectral-domain information improves robustness and transferability of learnt representations (Kara et al., 2024; Fu & Hu, 2025; Zhang et al., 2022a; Liu et al., 2024). However, existing PPG foundation models either focus solely on time-domain data or use frequency-based methods that leverage fixed-window Fourier transform and late fusion, limiting their ability to capture the adaptive, hierarchical time-frequency features of non-stationary physiological signals such as PPG (Chen et al., 2025; Masserano et al., 2025). To address these limitations, we introduce wavelet-based PPG foundation model, which explicitly learns cross-scale interactions in the time–frequency domain and enables broad generalization across diverse downstream tasks.

Physiological signals such as PPG are inherently multi-scale: local waveform morphology encodes vascular health, and long-term trends capture heart rate variability and global rhythm dynamics

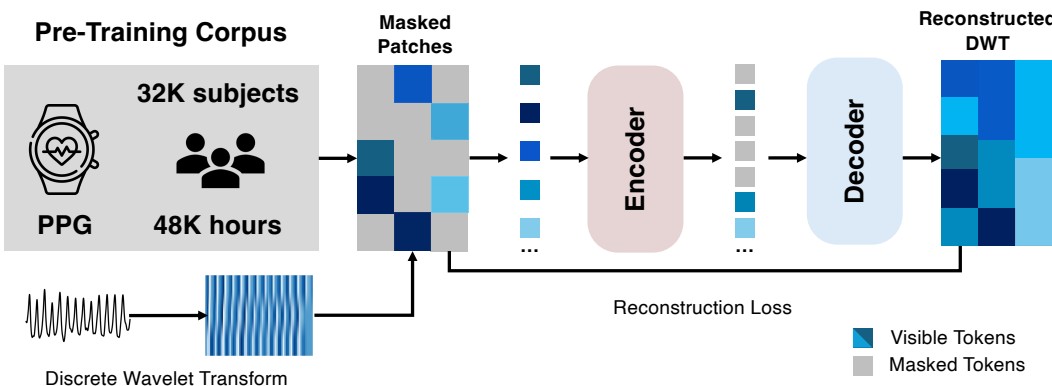

Figure 1: Masked Multiscale Reconstruction for Photoplethysmography (PPG) signals.

(Charlton et al., 2022b; Namvari et al., 2022; Sherebrin & Sherebrin, 2002). Prior approaches for PPG foundation models—temporal embeddings, patient-aware contrastive learning (Abbaspourazad et al., 2024b), morphology-based objectives (Pillai et al., 2024), generally overlook this hierarchical multi-scale structure of PPG. Even hybrid time–frequency models (Zhang et al., 2022a) often treat these domains separately, limiting multi-scale representation learning—a capability crucial for downstream health-related tasks that require features spanning granularities from beat-level morphology to global temporal semantics.

Wavelet decomposition offers a natural, physiologically aligned approach for analyzing PPG in the time-frequency domain. By adaptively trading off time and frequency resolution, wavelets can help in capturing waveform changes, and short-term heart-rate variability in a unified, multi-resolution representation. Motivated by these insights, we establish the Masked Multiscale Reconstruction (MMR) framework: raw PPG signals are decomposed into multiple wavelet bands using the Discrete Wavelet Transform (Daubechies, 1992; Mallat, 2002), and the model is trained to reconstruct masked coefficients across scales. This approach encourages the learning of rich, cross-resolution embeddings.

To summarize, our contributions are: *(i)* We pretrain a large-scale *wavelet-based PPG foundation model* on ∼48K hours of real-world wearable PPG data with a masked multiscale reconstruction objective, enabling the model to capture rich time–frequency information across multiple scales. *(ii)* We demonstrate robust generalization across 19 diverse downstream tasks and provide detailed ablations that examine the impact of design choices such as wavelet family, decomposition scales, and patch size, further underscoring the promise of wavelets for building generalizable PPG foundation models.

## 2 RELATED WORK

Self-supervised pre-training has become the dominant paradigm for large-scale biosignal modeling. Prior PPG foundation models have leveraged large datasets to learn general representations, using contrastive learning or waveform-based objectives to improve generalization (Abbaspourazad et al., 2024b; Pillai et al., 2024; Saha et al., 2025). Multimodal foundation models transfer representations across PPG, ECG, and other signals, and masked reconstruction has shown promise for multivariate health time series (Abbaspourazad et al., 2024a; Yang et al., 2023; Narayanswamy et al., 2024; Xu et al., 2025). These works mark a shift from traditional task-specific models to general-purpose biosignal foundation models.

Prior frequency-aware models rely on fixed-window spectral features, limiting their ability to capture the adaptive, hierarchical, multi-scale rhythms inherent across time-frequency in physiological signals (Zhang et al., 2022b; Liu et al., 2023; Kara et al., 2024; Cheng et al., 2025; Fu & Hu, 2025; Duan et al., 2024). Early works have applied wavelets to PPG for denoising, feature extraction, or task-specific pipelines (Alafeef & Fraiwan, 2020; Singh et al., 2023; Shao et al., 2021), while more recent deep learning work has applied end-to-end wavelets for time-series tokenization (Masser-

ano et al., 2025) and modeling biosignals like ECG and EMG (Chen et al., 2025). These methods, however, do not explicitly model cross-resolution interactions at scale for large, real-world PPG datasets.

In contrast, our MMR framework leverages multi-resolution wavelet decomposition to pre-train a large-scale PPG foundation model. By reconstructing masked coefficients across scales, MMR explicitly captures hierarchical, physiologically grounded structure—from local waveform morphology to intermediate oscillations and long-term rhythms. This approach goes beyond time-only and fixed-resolution spectral methods, producing robust, transferable embeddings that generalize across diverse cardiovascular and health monitoring tasks. See Appendix A.3 for an extended related works.

## 3 METHOD

To build a Masked Multiscale Reconstruction model, we transform PPG signals into multiple time–frequency scales using the Discrete Wavelet Transform (DWT) The resulting wavelet coefficients are interpolated and stacked to form a 2D coefficient map, which is partitioned into patches, many of which are masked, and then processed by a Vision Transformer (ViT) (Dosovitskiy et al., 2020). The model is trained with a multi-scale reconstruction objective, aiming to recover the masked coefficients and thereby learn robust signal representations (Fig. 1).

**Discrete wavelet transform.** The discrete wavelet transform (DWT) decomposes a signal into an approximation $A_J$ and detail bands $D_j{}_{j=1}^{J}$ using paired low- and high-pass filters, with each level downsampled by half for joint time–frequency localization (Mallat, 2002; Daubechies, 1992). Unlike the Fourier transform (Bracewell, 1989), which represents global frequency content and has zero time resolution, or the Short-Time Fourier Transform (STFT) (Durak & Arikan, 2003), which uses fixed time windows, the DWT adapts across scales, offering wide temporal support for low-frequency components and sharp temporal localization for high-frequency transients (Masserano et al., 2025; Stephane, 1999). This makes wavelets well-suited for nonstationary signals with localized bursts or discontinuities. In the DWT, the approximation and detail coefficients are obtained by inner products with scaling and wavelet functions. The scaling function $(\phi_{J,k})$, which acts as a low-pass filter, is used to find the approximation coefficients $(a_k^J)$, while the wavelet function $(\psi_{j,k})$, which acts as a high-pass filter, is used for the detail coefficients $(d_k^j)$. The approximation coefficients capture the low-frequency (coarse) structure of the signal, while the detail coefficients capture the high-frequency (fine) variations, providing a multi-resolution representation (Daubechies, 1992).

To obtain DWT coefficients from PPG signals, we first performed an empirical ablation study (Section 5.4) at a smaller data/model scale over multiple wavelet families and decomposition levels, and found the Haar wavelet (Haar, 1909) with level-3–4 decompositions to perform reliably across multiple downstream tasks. For our full-scale training, we employ a level-3 Haar DWT implemented with `PyWavelets` (Lee et al., 2019). For the Haar wavelet, approximation coefficients correspond to local averages, whereas detail coefficients capture signed differences, summarizing coarse trends while isolating localized variations. We interpolate the detail and approximation coefficients obtained from the DWT at each subband level using zero-order 1D interpolation, stretching the coefficients at level $j$ to match the original signal length. The resulting subbands are then concatenated in decreasing order of frequency, with high-frequency detail coefficients stacked at the top and the low frequency approximation band at the bottom, forming a 2D map of shape $[n_{\text{subbands}}, T]$, where $T$ is the segment length. Further details on construction of the 2D coefficient maps are provided in App. A.6.1.

**Masked Multiscale Reconstruction – MMR.** We adopt a Vision Transformer (ViT) encoder–decoder within the masked autoencoder framework (He et al., 2022). The 2-D wavelet coefficient map is divided into non-overlapping patches of size $(1, 25)$ along the temporal axis, yielding a token sequence $\{x_p\}_{p\in\mathcal{P}}$ across subbands. Fixed 2-D sine–cosine positional embeddings encode both temporal order and subband index. During pretraining, a random subset $\mathcal{M} \subset \mathcal{P}$ of 75% patches is masked, and the decoder reconstructs the missing coefficients from the visible tokens $\mathcal{P} \setminus \mathcal{M}$. Our MMR objective exploits the hierarchical structure of the DWT: coarse approximation bands $A_J$ encode global trends at scale $2^J$, while detail bands $\{D_j\}$ refine these trends at progressively

higher frequencies. The reconstruction task therefore encourages the encoder to capture both *top-down* (coarse → fine) and *bottom-up* (fine → coarse) dependencies, rather than treating each band independently. This approach encourages cross-scale feature sharing reminiscent of classical multiresolution analysis (Mallat, 2002), but learned directly by the Transformer. Formally, let $X_p \in \mathbb{R}^d$ denote the original coefficient vector in patch $p$ and $\hat{X}_p$ the reconstruction. The MMR loss is the mean-squared error over masked patches:

$$\mathcal{L}_{\text{MMR}} = \frac{1}{|\mathcal{M}|} \sum_{p \in \mathcal{M}} \left\| \hat{X}_p - X_p \right\|_2^2.$$

Equivalently, writing $X = \{A_J, D_1, \ldots, D_J\}$ for the multiscale decomposition,

$$\mathcal{L}_{\text{MMR}} = \frac{1}{|\mathcal{M}|} \left[ \sum_{j=1}^{J} \sum_{p \in \mathcal{M} \cap D_j} \left\| \hat{X}_p^{(j)} - X_p^{(j)} \right\|_2^2 + \sum_{p \in \mathcal{M} \cap A_J} \left\| \hat{X}_p^{(A_J)} - X_p^{(A_J)} \right\|_2^2 \right],$$

which makes explicit that errors are penalized across all scales. This multiscale masking compels the encoder to jointly model spectral and temporal dependencies, yielding richer latent features for downstream tasks.

## 4 EXPERIMENTAL SETTING

Here, we describe the datasets, pretraining setup, and evaluation protocols used to evaluate the MMR approach.

**Dataset** We pretrain our encoder on large-scale PPG segments collected from diverse REDACTED smartwatches. Each input is a 10-second segment, chosen as a practical trade-off between signal quality and model input length: shorter windows reduce the likelihood of motion artifacts or device dropout, while still capturing multiple cardiac cycles necessary for reliable waveform representation. The majority of segments were sampled at lower frequency range of 25 Hz-100Hz, reflecting the low-power, battery-constrained acquisition typical of free-living wearable devices. This setting is deliberately more challenging than those used in much of the prior work, which often relies on clean clinical signals or generic time-series datasets (Pillai et al., 2024). By contrast, our pretraining data reflect the noise, variability, and resource constraints of real-world wearables, aligning model development directly with deployment conditions.

**Preprocessing** Before feeding each PPG segment as input to MMR, each segment is bandpass filtered (Liang et al., 2018; Lapitan et al., 2024), and subsequently z-score normalized to standardize amplitude across devices and users. These steps reduce trivial sources of variability so that the encoder focuses on physiologically meaningful structure. Because our signals come from real-world smartwatch deployments, they inevitably contain motion artifacts, poor skin contact, and environmental noise. Rather than eliminating this variability altogether, we apply a lightweight signal quality index (SQI)–based filter to discard only the most corrupted segments. We combine entropy (to capture waveform regularity) and autocorrelation (to assess periodicity), two metrics commonly used to identify usable cardiac waveforms (Elgendi, 2016; Karlen et al., 2012; Pradhan et al., 2017). This approach strikes a balance: the retained segments still reflect the variability of naturalistic conditions, but avoid extreme outliers that would otherwise overwhelm representation learning.

**Downstream Tasks** We evaluate the learned representations on 19 health-related downstream tasks spanning both classification and regression. Classification tasks include hypertension detection evaluated in both controlled laboratory protocols and free-living settings. We also assess premature ventricular contraction (PVC) detection, which provides clinically relevant information for arrhythmia monitoring and supports atrial fibrillation detection. In addition, we evaluate classification of laboratory biomarker abnormalities, including electrolytes (sodium, potassium, carbon dioxide), hematological indices (hemoglobin, white blood cells), and metabolic markers (creatinine; ). We also add two demographic tasks, age classification and age regression; see App. A.7 for details. To probe whether the representations capture sleep-related physiology, we further include four sleep-stage classification tasks (Wake, Light, Deep, REM) derived from wrist-worn PPG on DREAMT

Table 1: Comparison with Self-Supervised Baselines. Best downstream evaluation scores are shown in **bold**, second-best are underlined, and the [min–max] range across the five cross-validation test splits is reported in gray brackets.

| Classification - AUROC (↑) | SimCLR (5M) | MSN (2.5M) | TF-C (10M) | MTR (7M) | MMR (7M) | MMR-Light (2M) |
|---|---|---|---|---|---|---|
| **Age** | 70.59 [69.1 – 71.2] | 72.16 [71.1 – 72.9] | 72.03 [71.3 – 72.5] | 75.15 [74.3 – 76.3] | **78.15** [77.6 – 79.3] | 75.63 [75.1– 76.6] |
| Hypertension - Lab | 69.58 [61.2 – 76.6] | 66.90 [53.6 – 76.6] | 73.10 [63.0 – 90.9] | **77.53**[64.0 – 90.1] | 75.34 [58.9 – 95.1] | 76.90 [60.1 – 92.2] |
| Hypertension - Free Living | 60.93 [59.1 – 62.8] | 55.16 [53.6 – 56.3] | 62.92 [61.3 – 65.0] | 67.01 [66.3 – 69.3] | 69.55 [68.0 – 71.1] | 68.43 [65.9 – 69.3] |
| PVC | 74.43 [67.9 – 82.4] | 73.60 [65.7 – 79.1] | 70.25 [64.9 – 74.8] | 83.72 [74.5 – 90.7] | **84.62** [74.5 – 90.7] | 83.67 [75.0 – 90.1] |
| Carbon Dioxide | **65.07** [53.5 – 83.7] | 62.90 [51.9 – 78.3] | 63.02 [53.1 – 77.2] | 65.01 [47.1 – 78.2] | 64.24 [44.1 – 76.1] | **65.07** [46.6 – 76.6] |
| Creatinine | 54.18 [38.8 – 71.4] | 56.45 [45.6 – 70.4] | 53.08 [47.7 – 63.8] | 53.49 [39.1 – 73.0] | **56.45** [45.2 – 74.1] | 54.97 [39.3 – 72.9] |
| Hemoglobin | **56.10** [29.1 – 69.8] | 52.04 [35.4 – 74.2] | 51.55 [40.2 – 64.3] | 53.19 [33.8 – 76.5] | 54.59 [38.1 – 77.1] | 52.51 [31.2 – 82.6] |
| Potassium | 71.63 [62.3 – 85.1] | 69.47 [58.8 – 80.8] | 73.76 [59.9 – 81.2] | **74.66** [65.4 – 87.3] | 73.68 [51.2 – 88.3] | 74.38 [52.4 – 89.1] |
| Sodium | 60.37 [53.5 – 65.7] | 60.39 [49.6 – 72.7] | 55.18 [41.7 – 71.6] | 59.08 [45.1 – 66.3] | **64.86** [57.2 – 73.4] | 60.64 [50.1 – 65.1] |
| White Blood Cells | **79.51** [55.9 – 86.0] | 76.93 [49.9 – 88.3] | 77.37 [48.2 – 86.3] | 75.69 [39.0 – 91.6] | 75.91 [42.4 – 92.6] | 75.65 [40.7 – 91.3] |
| **Wake** | 62.91 [62.6 – 63.2] | 64.25 [64.0 – 64.5] | 63.71 [63.5 – 64.1] | 64.61 [64.3 – 64.9] | **65.88** [65.6 – 66.2] | 65.30 [65.0 – 65.6] |
| **Light** | 56.11 [55.8 – 56.3] | 55.79 [55.5 – 56.0] | 54.97 [56.1 – 57.4] | 56.14 [55.8 – 56.4] | **57.13** [56.8 – 57.4] | 56.65 [56.4 – 56.9] |
| **Deep** | 54.64 [53.9 – 55.3] | 54.36 [53.5 – 55.1] | **58.19** [56.1 – 57.4] | 56.94 [56.1 – 57.4] | 57.71 [56.9 – 58.3] | 57.46 [56.4 – 58.9] |
| **Rem** | 53.91 [53.52 – 54.3] | 54.89 [54.4 – 55.2] | 54.46 [54.3 – 55.0] | 54.23 [53.8 – 54.6] | **55.66** [55.2 – 56.0] | 54.45 [54.0 – 54.8] |
| **Average** | 63.57 ± 8.08 | 62.55 ± 8.07 | 63.92 ± 9.38 | 65.46 ± 9.95 | **66.70** ± 9.35 | 65.84 ± 9.71 |
| **Regression - MAE (↓)** | | | | | | |
| **Age** | 9.39 [9.2 – 9.5] | 9.05 [8.9 – 9.2] | 8.96 [8.9 – 9.0] | 8.77 [8.6 – 8.8] | **8.37** [[8.2 – 8.5 ] | 8.69 [8.5 – 8.8] |
| Sys. BP (Lab) | 11.02 [9.0 – 12.0] | 11.02 [8.9 – 11.8] | **10.93** [9.1 – 11.6] | 11.30 [9.7 – 11.8 | 11.28 [9.4 – 11.8] | 11.22 [9.5 – 11.8] |
| Dias. BP (Lab) | 7.95 [6.8 – 10.1] | 8.07 [6.7 – 10.2] | 7.95 [7.2 – 10.2] | 7.96 [7.2 – 10.2] | 7.76 [6.8 – 10.1] | **7.75** [6.6 – 10.1] |
| Sys. BP | 11.75 [11.6 – 11.8] | 11.82 [11.7 – 11.9] | 11.75 [11.6 – 11.8] | 11.65 [11.5 – 11.7] | **11.61** [11.5 – 11.7] | 11.63 [11.5 – 11.8] |
| Dias. BP | 9.47 [9.2 – 9.7] | 9.52 [9.2 – 9.8] | 9.45 [9.2 – 9.7] | 9.41 [9.1 – 9.5] | **9.32** [9.1 – 9.4] | 9.36 [9.1 – 9.6] |
| **Average** | 9.92 ± 1.34 | 9.90 ± 1.35 | 9.81 ± 1.37 | 9.82 ± 1.43 | **9.67** ± 1.54 | 9.73 ± 1.48 |

dataset (Wang et al., 2024b;a; Goldberger et al., 2000), which are evaluated in a one-vs-rest setting. Regression tasks focus on systolic and diastolic blood pressure prediction, capturing continuous cardiovascular physiology.

**Baselines** We compare MMR against four groups of baselines. First, we add a masked time reconstruction baseline, a time-domain masked autoencoder that operates directly on patchified raw PPG and reconstructs the waveform. ( A.5.1) Second, self-supervised learning methods trained on the same wearable PPG data, including SimCLR (Chen et al., 2020), Masked Siamese Network (MSN) (Assran et al., 2022), and TF-C (Zhang et al., 2022a), representing contrastive, masked patch/data matching, and multi-view frequency–time approaches, respectively (see Appendix A.6.3 for details). These baselines allow us to assess the benefit of our proposed architecture and pretraining strategy when applied to the same real-world PPG signals. Third, we evaluate open-source pretrained models, divided into two categories: (i) general-purpose time-series foundation models such as Chronos and Chronos-Bolt (See Appendix A.5.2 ) (Ansari et al., 2024), which are trained on diverse multivariate time-series but not specifically on physiological signals, and (ii) domain-specific models such as PaPaGei (Pillai et al., 2024), which leverage high-quality fingertip PPG with stratified binning and explicit morphological feature learning to capture clinical waveform properties at high sampling rates (125–500 Hz); we use the PaPaGei-S variant in this work. These models provide a point of comparison to evaluate whether pretraining on clean or general-purpose data transfers effectively to noisy, wearable PPG. We also include a baseline, PaPaGei-Ours, which is trained on our pretraining corpus while following the architectural design and training protocol of PaPaGei. Finally, we include classical statistical feature–based models without pretraining to provide a non-learned baseline. Together, these baselines span the current state of both general-purpose and domain-specialized representation learning for PPG signals, highlighting the advantages of MMR in capturing real-world cardiovascular physiology.

## 5 DOWNSTREAM EVALUATION

We evaluate the quality and generalization of the learned representations by training downstream classifiers on top of frozen encoders. We consider both our full MMR model ( 7M params )and a smaller variant, MMR-LIGHT (2M params), which has fewer parameters to study efficiency-performance trade-offs. For all experiments, encoder representations serve as input features to ran-

dom forest models for classification and regression tasks, with performance measured on held-out test data using 5-fold cross-validation (More evaluation details in A.6.4. Binary classification tasks are evaluated via AUROC, and regression tasks via mean absolute error (MAE). Our evaluation encompasses several aspects: (i) performance across 19 downstream health-related tasks compared to self-supervised and pretrained baselines, (ii) analysis of the learned embedding space to assess patient discriminability and physiological structure, (iii) the impact of pretraining data size and model parameter scaling, and (iv) ablation studies examining key architectural choices such as wavelet family, decomposition level, and patch size. This comprehensive evaluation allows us to assess both the predictive power and the interpretability of the representations learned by MMR and MMR-Light.

## 5.1 Evaluating Transferability of Learned Features Across Diverse Downstream Tasks

Detailed results for all baselines are reported in Tables 8 and 9. Across a broad suite of 19 downstream health-related tasks, MMR consistently achieves competitive performance relative to state-of-the-art baselines. On average, MMR reaches 66.70% AUROC across 14 binary classification tasks spanning cardiovascular outcomes, lab abnormalities, demographic tasks, and sleep-stage recognition, with notable strengths in PVC detection ( 84%) and free-living hypertension classification. Its lightweight variant, MMR-Light, achieves comparable performance, demonstrating that parameter efficiency can be maintained without substantial loss in predictive quality. For regression tasks, MMR attains the lowest error on diastolic blood pressure in the lab setting and on both hypertension-related regression tasks in free-living conditions. MMR achieves strong performance on both demographic tasks set up on our evaluation cohort: it achieves (78.15%) AUROC for age classification, and attains the MAE of 8.36 for age regression.

These results hold even when compared to self-supervised baselines trained on identical wearable PPG data (SimCLR, MSN, TF-C). While TF-C, leveraging both time- and frequency-domain augmentations, performs well on hypertension and systolic BP regression, SimCLR and MSN underperform on free-living hypertension and PVC detection. The closest competitve baseline is the masked autoencoder baseline (MTR), which shares the same backbone and pretraining data as MMR but optimizes a purely temporal reconstruction objective, illustrating the strength of reconstruction-based training for PPG. Nevertheless, in most classification and regression tasks MMR improves over MTR suggesting that introducing an explicit multi-resolution wavelet-based inductive bias into large-scale pretraining yields more informative and transferable PPG representations (see Appendix A.5.1).

When compared to pretrained PPG models, MMR outperforms PaPaGei and PaPaGei-Own approximately by 5% and 3% average AUROC across classification tasks. PaPaGei, trained on high-quality fingertip PPG at high sampling rates, relies on fiducial point extraction, which can limit adaptability to noisy, low-sampling-rate wearable data. Similarly, MMR improves over the large, general-purpose time-series model Chronos (10 of 14 tasks; average +4.5% AUROC), showing that domain-specific pretraining on large-scale wearable PPG is critical for capturing cardiovascular signal characteristics that general multivariate models may miss.

Taken together, these results suggest that MMR produces useful representations that show strong performance across a range of downstream tasks.

## 5.2 Learned Feature Analysis

To further interpret how MMR captures clinically relevant information, we examine the structure of the learned embeddings on an unseen downstream task: the Hypertension- Free Living dataset.

**Discriminability of Patient-Wise Embeddings.** To evaluate the quality and interpretability of the learned representations, we examine the separation of patient embeddings by computing inter-patient distances. Well-separated embeddings indicate that the model separates diverse patient cohorts, which can improve robustness and accuracy in downstream tasks Pillai et al. (2024). Figure 2a shows the average pairwise distances across patients in the Hypertension- Free Living dataset. MMR achieves the largest separation among all methods, demonstrating its ability to disentangle individual patient identities in the embedding space. In contrast, contrastive baselines such as SimCLR and

Table 2: Comparison with Open-source Pretrained Models and Statistical Features. Best downstream evaluation scores are shown in **bold**, second-best are underlined, and the minimum–maximum range across the five cross-validation splits is reported in gray brackets.

| Classification - AUROC (↑) | Stat. Feat. | Chronos (200M) | PaPaGei (5M) | PaPaGei-Ours (5M) | MMR (7M) | MMR-Light (2M) |
|---|---|---|---|---|---|---|
| **Age** | 60.31 [59.4 − 62.0] | 71.09 [70.8 − 71.3] | 67.19 [66.6 − 67.6] | 68.14 [67.5 − 69.5] | **78.15** [77.6 − 79.3] | 75.63 [75.1 − 76.6] |
| Hypertension - Lab | 69.28 [59.7−83.1] | 67.26 [55.8−79.8] | 65.85 [58.6−75.4] | 66.08 [58.2−82.6] | 75.34 [58.9 − 95.1] | **76.90** [60.1 − 92.2] |
| Hypertension - Free Living | 56.59 [54.7−57.7] | 59.95 [58.3−61.8] | 59.94 [59.2−60.2] | 63.61 [62.5−64.8] | **69.55** [68.0 − 71.1] | 68.43 [65.9 − 69.3] |
| PVC | 70.41 [64.2−77.2] | 65.73 [65.3−75.3] | 72.20 [67.0−79.9] | 74.59 [69.1−81.9] | **84.62** [74.5 − 90.7] | 83.67 [75.0 − 90.1] |
| Carbon Dioxide | 57.20 [50.3−69.2] | 61.14 [54.0−75.1] | 61.23 [52.3−75.3] | 63.45 [52.7−79.8] | 64.24 [44.1 − 76.1] | **65.07** [46.6 − 76.6] |
| Creatinine | 49.32 [40.2−56.7] | **61.14** [49.2−74.0] | 51.86 [43.4−70.0] | 54.22 [41.7−76.7] | 56.45 [45.2 − 74.1] | 54.97 [39.3 − 72.9] |
| Hemoglobin | 56.13 [35.6−65.1] | 53.65 [43.7−59.3] | 49.28 [39.5−59.2] | **60.32** [32.5−76.1] | 54.59 [38.1 − 77.1] | 52.51 [31.2 − 82.6] |
| Potassium | 47.87 [41.6−55.3] | 63.49 [55.4−85.1] | 68.51 [56.1−77.2] | 64.91 [48.1−70.83] | 73.68 [51.2 − 88.3] | **74.38** [52.4 − 89.1] |
| Sodium | 52.72 [47.5−56.7] | **63.33** [53.6−74.2] | 58.44 [51.1−70.3] | 59.41 [56.1−63.5] | 64.86 [57.2 − 73.4] | 60.64 [50.1 − 65.1] |
| White Blood Cells | 55.52 [52.7−58.3] | **77.51** [55.4−88.1] | 73.64 [44.1−87.1] | 74.71 [52.7−84.8] | 75.91 [42.4 − 92.6] | 75.65 [40.7 − 91.3] |
| **Wake** | 61.56 [61.2 − 61.9] | 61.63 [61.3 − 61.9] | 63.03 [62.7 − 63.4] | 61.95 [61.6 − 62.3] | **65.88** [65.6 − 66.2] | 65.30 [65.0 − 65.6] |
| **Light** | 55.76 [55.5 − 56.0] | 55.75 [55.4 − 56.0] | **57.15** [56.9 − 57.4] | 56.08 [55.8 − 56.4] | 57.13 [56.8 − 57.4] | 56.65 [56.4 − 56.9] |
| **Deep** | 55.37 [54.6 − 56.1] | 55.43 [54.7 − 56.1] | **61.50** [60.9 − 62.2] | 54.21 [53.5 − 54.9] | 57.71 [56.9 − 58.3] | 57.46 [56.4 − 58.9] |
| **Rem** | 53.90 [53.6 − 54.4] | 53.78 [53.3 − 54.2] | 54.03 [53.6 − 54.4] | 54.05 [53.6 − 54.9] | 55.66 [55.2 − 56.0] | 54.45 [54.0 − 54.8] |
| Average | 57.28 ± 6.20 | 62.21 ± 6.51 | 61.03 ± 7.33 | 63.22 ± 6.12 | **66.70** ± 9.35 | 65.84 ± 9.71 |
| **Regression - MAE (↓)** | | | | | | |
| **Age** | 9.96 [9.8 − 10.1] | 9.30 [9.2 − 9.4] | 9.72 [9.6 − 9.8] | 9.47 [9.3 − 9.6] | **8.37** [8.2 − 8.5] | 8.69 [8.5 − 8.8] |
| Sys. BP (Lab) | 11.08 [9.3−11.9] | **10.79** [9.2−11.9] | 11.04 [9.6−12.2] | 11.28 [9.8−12.3] | 11.12 [9.4−11.7] | 11.07 [9.4−11.8] |
| Dias. BP (Lab) | 8.12 [6.7−10.3] | 7.93 [7.0−8.9]a | 8.09 [6.9−10.1] | 7.98 [6.9−9.9] | 7.90 [6.9−10.1] | **7.88** [6.9−10.1] |
| Sys. BP | 11.75 [11.6−11.8] | 11.82 [11.7−11.9] | 11.75 [11.6−11.8] | 11.74 [11.6−11.8] | **11.63** [11.4−11.7] | 11.66 [11.5−11.8] |
| Dias. BP | 9.47 [9.2−9.7] | 9.52 [9.2−9.8] | 9.45 [9.2−9.7] | 9.45 [9.2−9.7] | **9.36** [9.1−9.5] | 9.37 [9.1−9.6] |
| Average | 10.08 ± 1.27 | 9.87 ± 1.33 | 10.01 ± 1.28 | 9.98 ± 1.37 | **9.67** ± 1.54 | 9.73 ± 1.48 |

TF-C show weaker separation, with many embeddings collapsing together, while PaPaGei performs moderately well but still lags behind MMR.

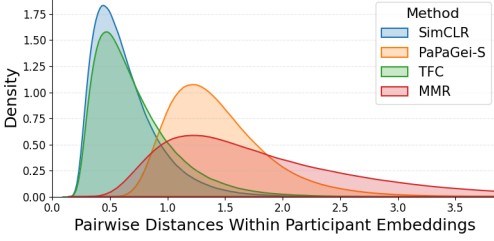 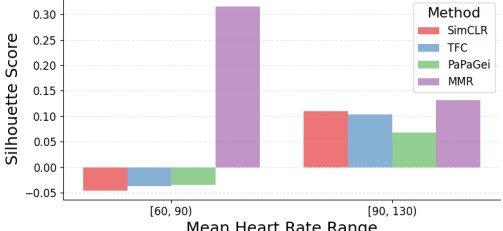

(a) Comparison of participant-wise distance distributions of learned embeddings across different baselines. Wider curves mean stronger separation and discrimination.

(b) Silhouette scores of heart rate clusters in the average patient embeddings learned by different baselines. A higher score indicates better-defined clusters and stronger separation in the embedding space.

Figure 2: Analyzing the learned embeddings on the unseen users in Hypertension–Free Living data.

**Physiological Structure in the Embedding Space.** Beyond patient-wise discriminability, the learned embeddings capture meaningful physiological variation. Visualizing participant-level mean embeddings with t-SNE (van der Maaten & Hinton, 2008), colored by heart rate ranges (Figure 13), reveals a smooth gradient from low to high heart rate. This observation indicates that MMR encodes underlying cardiovascular dynamics rather than producing arbitrary or randomly aligned representations. To quantify this structure, we compute silhouette scores (Rousseeuw, 1987) for elevated (90-130 bpm) vs. normal (60-90 bpm) heart rate groups. MMR consistently achieves the highest scores, demonstrating that its embeddings cluster more coherently by physiological state than those of competing baselines. These findings suggest that MMR produces *physiologically aligned representations*, enabling downstream models to leverage subtle but clinically relevant variations in cardiovascular signals.

## 5.3 Evaluating MMR at Varied Data and Parameter Settings

We next examine how pretraining dataset size and model capacity influence downstream performance. By comparing MMR across different amounts of pretraining data and varying parameter scales, we gain insight into the trade-offs between model complexity, data efficiency, and predictive accuracy.

**Pretraining Data Scaling** We pretrain MMR (7M) on datasets of increasing size, 1M, 5M, and 17M segments, to examine how the amount of pretraining data affects downstream classification performance. As shown in Fig. 3, on average larger pretraining corpus leads to increase in overall mean AUROC across key downstream evaluation tasks (More details in Appendix A.9). The App. table 7 shows larger datasets consistently improve results on hypertension and PVC detection, highlighting the benefit of data volume. The smallest dataset (1M segments) underperforms relative to larger splits, achieving 68.0% and 66.1% AUROC on lab and free-living hypertension tasks, compared to 69.1% and 67.5% for 5M segments. Pretraining on the full 17M segments yields the strongest overall performance across tasks. Some lab biomarkers, such as abnormal WBC and creatinine, show little improvement with additional data, suggesting that certain endpoints are less sensitive to segment-level diversity. Overall, these results indicate that diverse and large-scale pretraining data—spanning multiple users, devices, and real-world contexts—is critical for learning robust, generalizable representations.

**Model Scaling** We next examine how model capacity affects downstream performance, holding the pretraining set fixed at 17 Million segments. As shown in Fig 3, on average larger model parameter count helps with downstream performance. In App. Table Table 6), we note that the mid-sized 7M-parameter model also performs strongly across tasks, ranking first or second on 7 of 9 classification tasks. The smaller 2M model achieves competitive results, indicating that MMR-Light provides an effective, parameter-efficient alternative suitable for

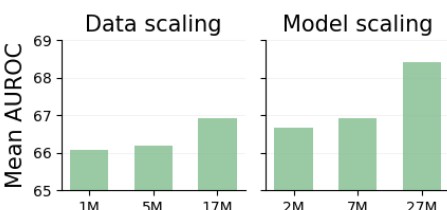

Figure 3: Mean AUROC on key tasks across data and model scales.

deployment on resource-constrained devices. Larger models offer notable improvements for select endpoints, such as +10% AUROC on hypertension-lab and +5% on sodium prediction, but these gains do not generalize uniformly across all tasks. Overall, scaling model capacity can enhance performance for specific tasks, but even compact models retain substantial predictive power.

## 5.4 Ablation Studies

To understand the contribution of individual components within MMR, we conducted ablation experiments using a 2M-parameter model (MMR-Light) pretrained on a 1M-segment subset. We systematically varied the wavelet family, decomposition level, masking strategy, and patch size, and evaluated the resulting impact across all classification tasks. In Figure 4, the Hypertension-Free Living dataset is labeled as BP, and the Hypertension Lab as BP Lab.

**Wavelet family design.** We evaluated several wavelet bases (db4, bior2.2, bior4.4, Haar) (Daubechies, 1988; Karoui & Vaillancourt, 1994; Haar, 1909) under identical training conditions (2M parameters, 1M segments). Across hypertension classification tasks, Haar consistently achieved the highest AUROC, and it also outperformed most other wavelets on PVC detection. This advantage likely stems from Haar's compact and sharply changing basis functions, which are well-suited to capturing abrupt waveform changes that signal irregular heartbeats (Yang et al., 2019). In contrast, smoother wavelets like db4 and bior4.4 can miss these fine transients, reducing performance on tasks that depend on detecting sharp signal features. For other biomarkers and vitals, differences across wavelets were smaller, with average AUROC generally between 64–68. Notably, bior2.2 improved Creatinine prediction, while bior4.4 gave modest gains for WBC and Sodium lab outcomes. These results suggest that smoother wavelets may be preferable for tasks dominated by slower, more global signal dynamics, whereas Haar supports tasks requiring the detection of rapid, localized changes.

**Effect of decomposition depth.** We evaluated the effect of wavelet decomposition depth (levels 2–5 using Haar) in our PPG → DWT transformation on a smaller setting, using a model pretrained on a 1M-sample subset where more exhaustive sweeps are feasible. Decomposition depth determines the number of multi-resolution sub-bands used to represent the signal. Moderate depths (levels 3) consistently yielded the best performance on hypertension classification, while deeper decompositions (level 5) led to a 7–8 % AUROC drop, likely because excessive decomposition fragments the signal and discards fine-grained, predictive features. A similar pattern was observed for PVC detection: level 3 achieved over 80% AUROC with only 1M pretraining samples, outperforming level 2 (73%) and level 5 (75%). Some lab biomarkers, including WBC, Sodium, and carbon dioxide, benefited from deeper decomposition, indicating that tasks dominated by slower, global signal trends may require more sub-bands. Overall, these results suggest that decomposition depth can have a task-dependent impact, highlighting the potential value of adaptive decomposition strategies for future work.

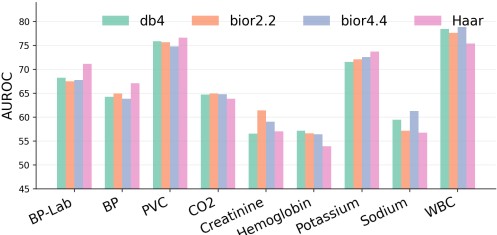 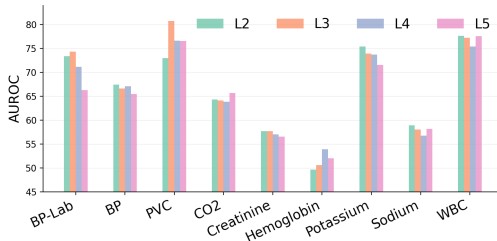

(a) Comparison across different wavelet families (db4, bior2.2, bior4.4, and Haar) shows that Haar provides consistently strong performance across tasks.

(b) Evaluation of decomposition levels (L2–L5) indicates that moderate depths, particularly L4, achieve the best balance.

Figure 4: Ablation of various wavelet families and decomposition levels for PPG signal analysis.

**Patch size ablation** Building on the previous analysis of decomposition depth, we next examine how patch size—the temporal resolution of input segments—affects downstream performance (refer Fig. 7b in Appendix). Patch size determines how the encoder captures local transients versus broader signal dynamics. Smaller patches, such as $(1, 25)$, consistently achieve the strongest hypertension classification, highlighting the importance of preserving fine-grained temporal details. Larger patches, like $(1, 100)$, tend to smooth out fine-grained events, leading to a 6–7% drop in hypertension lab AUROC, but provide modest gains for lab biomarkers such as WBC and potassium. The intermediate patch size $(1, 50)$ offers a compromise, performing reasonably on lab outcomes but remaining weaker for hypertension. The across-band scheme $(2, 25)$ underperforms across nearly all tasks, suggesting that mixing sub-bands can reduce discriminative power.

Overall, these results reveal that different tasks benefit from distinct temporal and frequency scales and suggest that multi-scale PPG representations provide complementary cues, highlighting wavelet-based encodings as an effective pretraining strategy for robust and adaptive physiological monitoring.

## 6 DISCUSSION AND FUTURE WORK

Foundation models for biosignals hold strong promise for generalizable digital health applications, yet most existing approaches overlook the spectral structure underlying physiological rhythms. In this work, we introduced Masked Multiscale Reconstruction (MMR), a pretraining framework for PPG that leverages wavelet-based time–frequency representations, and demonstrated robust performance across 19 diverse health tasks, matching or surpassing state-of-the-art baselines. Our analyses show that MMR embeddings capture physiologically meaningful information and enable subject-level discrimination, while ablations highlight the importance of explicitly modeling spectral hierarchies for representation learning. One promising direction for further improving large-scale wavelet-based biosignal pretraining is to develop adaptive multiscale decomposition strategies and architectures that more effectively exploit wavelet-domain representations. More broadly, our findings suggest that learning directly in the joint time–frequency domain is a powerful paradigm for

PPG foundation models, opening paths toward multimodal integration, longitudinal modeling, and clinically meaningful health prediction.

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

# A APPENDIX

## A.1 REPRODUCIBILITY STATEMENT

Due to data-use agreements and internal policies restrictions, we are unable to release the full source code or pretraining data. However, to support reproducibility and situate our results in a broader context, we provide a complete specification of the MMR architectures (MMR and MMR-Light)—including encoder/decoder depth, hidden dimensions, masking ratio, optimizer, and learning-rate schedule—in Appendix Section A.6.2. We further detail our preprocessing pipeline and discrete wavelet configuration (wavelet family, decomposition level, etc.) in Appendix Section A.6.1. We also include all the hyperparameter config and architectures used in our baseline methods in Section A.6.3.

## A.2 ETHICS STATEMENT

Our work uses physiological and behavioral data collected from consumer wearable devices. Because these signals can reveal sensitive information about individuals' health, habits, and daily routines, we applied strict safeguards for data governance, privacy, and fairness throughout all stages of the project. All datasets were obtained under informed consent. Participants agreed that sensor-derived information (e.g., steps, heart rate, sleep metrics, and photoplethysmography [PPG] signals) could be used in health-related research, for the development of new algorithms and features, and for publication in aggregated form. Consent was obtained via paper or electronic forms that clearly described the scope of data use, and explained that participation was voluntary and could be discontinued at any time. Personally identifying fields were removed before analysis, and only de-identified data were used for model development and evaluation.

Data used for downstream tasks were drawn from a mix of institutional review board (IRB)–approved internal studies and open datasets. Each contributing study underwent ethical review at the hosting institution or was classified as exempt when only de-identified records were involved. The downstream benchmarks include arrhythmia detection using paired ECG and PPG recordings to identify premature ventricular contractions (PVCs) with ECG-based labels verified by both automated and manual review, hypertension classification based on wrist PPG signals paired with reference blood pressure measurements collected under IRB-approved clinical protocols, and abnormal laboratory value prediction using PPG recordings linked to clinical biomarkers under protocols permitting secondary analysis of de-identified records.

## A.3 EXTENDED RELATED WORK

Self-supervised pretraining has emerged as the dominant paradigm for large-scale biosignal modeling. For example, (Abbaspourazad et al., 2024b) trained foundation models on PPG and ECG from ~141K Apple Watch users, demonstrating the value of contrastive learning at scale. In parallel, (Pillai et al., 2024) introduced PaPaGei , an open-source PPG foundation model trained on 20M unlabeled fingertip PPG segments that explicitly leverages waveform morphology, while (Saha et al., 2025) developed Pulse-PPG using 100 days of field data from 120 participants, showing improved efficiency and generalizability. Beyond single-modality PPG, multimodal biosignal foundations transfer representations across ECG, PPG, and other signals either via knowledge distillation (Abbaspourazad et al., 2024a) or unified embeddings (Yang et al., 2023). Related work has also applied masked reconstruction on multivariate health time series, yielding strong generative and discriminative performance on tasks such as activity classification (Narayanswamy et al., 2024; Xu et al., 2025). Together, these advances reflect a shift from task-specific models to general-purpose foundation models for biosignals.

While these foundation models highlight the value of large-scale self-supervision, most treat signals purely in the time domain. A growing body of work shows that explicitly incorporating spectral information provides a powerful inductive bias for robust and transferable representations. For instance, Time-Frequency Consistency (Zhang et al., 2022b) proposed aligning time- and frequency-domain views via contrastive loss, while bioFAME (Liu et al., 2023) introduced a frequency-aware transformer encoder with multi-head spectral filters. Similarly, FreqMAE (Kara et al., 2024) leveraged temporal-shifting encoders to model spectral content in multimodal IoT data. More recent approaches, such as FAT (Cheng et al., 2025), FEI (Fu & Hu, 2025), and MF-CLR (Duan et al., 2024),

further illustrate how spectral modeling can enhance time-series representation learning. These findings suggest that frequency-aware pretraining can serve as a complementary approach to large-scale training for physiological signals such as PPG. However, most rely on a fixed-size Fourier transform and fail to capture multi-scale representations of PPG.

**Comparison to frequency-aware MAE variants.** FreqMAE (Kara et al., 2024) uses a Short-Time Fourier Transform (STFT) for time-frequency representation of signals from different modalities. bioFame (Liu et al., 2023) employs only a fixed-window Discrete Fourier Transform (DFT) for modelling the raw signal. Instead, we propose to represent input signals using a wavelet-based DWT hierarchy (Daubechies, 1992) in combination with a cross-resolution reconstruction objective. Prior approaches relying on fixed-window STFT/DFT decompositions suffer from inherent limitations due to their uniform time–frequency resolution: a single fixed analysis window constrains all frequency components to share the same temporal support, which leads to suboptimal modelling of short-lived high-frequency transients and coarse characterization of long-range low-frequency structure. This rigid resolution also amplifies the impact of the Fourier uncertainty trade-off (Oppenheim, 1999), preventing these representations from adapting to the nonstationary, multi-scale nature of many real-world signals. In contrast, the multiresolution properties of the DWT provide scale-dependent localization, adaptively allocating fine temporal resolution to high-frequency content and fine spectral resolution to low-frequency content, yielding richer and more discriminative representations for reconstruction-based pretraining. We also note that DWT-based signal representations have been used effectively for biosignals in prior work, but not in combination with a masked reconstruction objective (Ponsiglione et al., 2021; Spyridou & Hadjileontiadis, 2007).

## A.4 MMR Ablations

**MMR is downstream data efficient** Wavelet analysis provides a natural way to capture information at different temporal scales by decomposing signals into multi-resolution frequency bands. Earlier PPG studies applied discrete wavelet transforms (DWT) for denoising and handcrafted features, for example, in respiratory rate estimation (Alafeef & Fraiwan, 2020), hypertension and diabetes detection (Singh et al., 2023), and peak stabilization pipelines (Shao et al., 2021). More recently, deep learning models have incorporated wavelets end-to-end, such as wavelet-based tokenization for time-series foundation models (Masserano et al., 2025) and PhysioWave (Chen et al., 2025), which couples learned wavelet decompositions, frequency guided masking with Transformers for physiological signals such as ECG and EMG. Building on this line of work, we introduce a multi-resolution masked

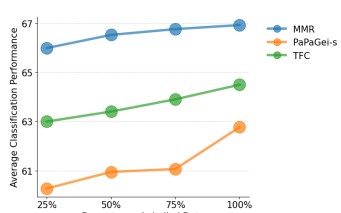

Figure 5: Average performance of MMR, PaPaGei-S, and TFC models across increasing percentages of labeled downstream data. Performance improves with more labeled data, with MMR consistently leading, followed by TFC and PaPaGei-S.

pretraining framework for large-scale PPG data collected from smartwatches in real-world settings. By leveraging the fact that health tasks rely on information at multiple signal granularities, our approach provides more physiologically grounded and transferable representations.

In this evaluation, the frozen MMR model representations were evaluated with varying proportions $\{25\%, 50\%, 75\%\}$ of labeled downstream data. MMR demonstrates clear efficiency in limited labeled settings, outperforming both TFC and PaPaGei-S. This early advantage persists as data availability increases, showing that MMR's representations remain robust and transferable across scales. TFC follows closely, improving steadily with additional supervision but never closing the gap with MMR. PaPaGei exhibits the steepest relative improvement as labeled supervision increases, making its performance at full data more competitive. Overall, these patterns highlight MMR's performance consistency across downstream data regimes.

**Masking strategy ablation.** We evaluate three masking strategies for our masked autoencoder: Random, Row-wise: masking entire rows of approximation or detail coefficients, and Cross-scale: masking columns across the hierarchical scales. Random masking achieves the strongest performance on hypertension classification, outperforming Row-wise by 5–6% and Cross-scale by 2–3%. This suggests that eliminating entire subband coefficients or masking across scales impedes the

model's ability to predict low-frequency coefficients without access to high-frequency components, and vice versa. Across most tasks, random masking either leads or performs competitively. However, task-specific patterns emerge: cross-scale proves advantageous for Creatinine and Sodium prediction. For WBC classification, both row-wise and cross-scale masking achieve 78%, representing a +3 point improvement over random. Overall, random masking serves as a strong default strategy, while structured masking (row-wise, cross-scale) offers selective benefits for particular biomarkers.

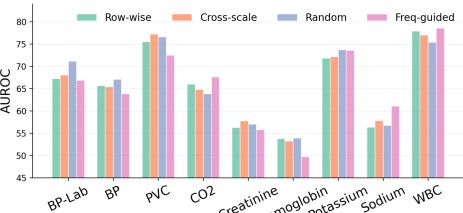 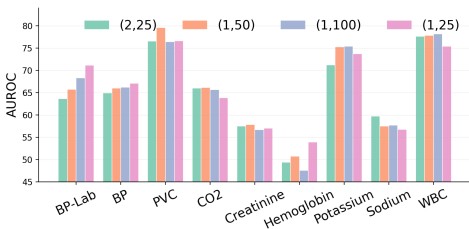

(a) Masking strategy: We ablate across four masking strategies: row-wise, cross-scale, random and frequency-guided, and find that random masking performs most consistently, providing a robust choice across downstream tasks.

(b) Patch size: We ablate across multiple patch size configurations, varying the number of sub-bands and temporal span per patch, and observe that $(1, 25)$ achieves the most consistent performance across the downstream task.

Figure 7: Ablation studies analyzing the effect of masking strategy and patch size in the proposed wavelet masked modeling framework, MMR.

**Preprocessing Ablation Experiments.**

We additionally stress-tested the normalization and interpolation choices used in our method. Our default configuration uses instance normalization with zero-order interpolation. We compared this to global normalization, as well as replacing zero-order interpolation with cubic and linear interpolations. As shown in Fig. 6, all four variants obtain very similar results across downstream tasks: the average score of our default setting is within 0.5–1.0 points of the alternatives, and the per-task spread between the best and worst variants are within 2-3 points. This indicates that our preprocessing approach is robust to the exact choice of normalization and interpolation scheme. In the figure, ars show average $\Delta$, and error bars show standard deviation over tasks.

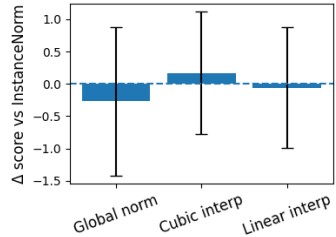

Figure 6: Mean performance difference of normalization/interpolation variants relative to our default setting across tasks.

## A.5    ADDITIONAL RESULTS DISCUSSION

### A.5.1    MASKED TIME VS MASKED MULTISCALE RECONSTRUCTION

To isolate the effect of the wavelet front-end, we introduce Masked Time Reconstruction (MTR) as a controlled baseline. MTR is a time-domain masked autoencoder that uses the same ViT encoder–decoder, masking ratio, patch size, optimizer, training schedule, and pretraining data as MMR, but operates on patchified raw PPG and reconstructs the waveform instead of DWT coefficients. As shown in Table 8, MTR is already a strong self-supervised learner and it generally outperforms the contrastive baselines (SimCLR, MSN, TF-C) in both AUROC and MAE, indicating that masked reconstruction of PPG is a competitive pretraining objective. However, MMR improves over MTR on 17 of 19 downstream tasks, with clear gains on key cardiovascular endpoints (e.g., free-living hypertension, PVC), demographics (age classification and regression), and sleep staging. Because architecture, data, and training are held strictly fixed, the observed gains stem from changing the representation and reconstruction target (time-domain waveform vs. multiscale DWT coefficients). This controlled ablation supports that the multiscale wavelet decomposition and masked reconstruction over DWT coefficients—is the key driver of the gains we observe for large-scale wearable PPG pretraining, beyond what an otherwise identical time-domain MAE can achieve.

Table 3: Comparison of Chronos, Chronos-Bolt, MMR, and MMR-Light on key downstream tasks.

| Classification - AUROC (↑) | Chronos (200M) | Chronos-Bolt (210M) | MMR (7M) | MMR-Light (2M) |
|---|---|---|---|---|
| **Age** | **71.09** [70.8 – 71.3] | **73.60** [72.1 – 74.1] | **78.15** [77.6 – 79.3] | 75.63 [75.1 – 76.6] |
| Hypertension - Lab | 67.26 [55.8 – 79.8] | 73.66 [67.5 – 85.6] | 75.34 [58.9 – 95.1] | 76.90 [60.1 – 92.2] |
| Hypertension - Free Living | 59.95 [58.3 – 61.8] | 66.92 [65.7 – 67.8] | 69.55 [68.0 – 71.1] | 68.43 [65.9 – 69.3] |
| PVC | 65.73 [63.5 – 75.3] | 75.23 [68.4 – 88.5] | **84.62** [74.5 – 90.7] | 83.67 [75.0 – 90.1] |
| Carbon Dioxide | 61.14 [54.0 – 75.1] | 63.16 [48.1 – 75.0] | 64.24 [44.1 – 76.1] | **65.07** [46.6 – 76.6] |
| Creatinine | 61.14 [49.2 – 74.0] | 60.88 [51.6 – 71.4] | **56.45** [45.2 – 74.1] | 54.97 [39.3 – 72.9] |
| Hemoglobin | 53.65 [43.7 – 59.3] | 57.80 [33.0 – 87.0] | 54.59 [38.1 – 77.1] | 52.51 [31.2 – 82.6] |
| Potassium | 63.49 [55.4 – 85.1] | 75.23 [68.5 – 88.5] | 73.68 [51.2 – 88.3] | **74.38** [52.4 – 89.1] |
| Sodium | 63.33 [53.6 – 74.2] | 61.33 [45.1 – 68.7] | **64.86** [57.2 – 73.4] | 60.64 [50.1 – 65.1] |
| White Blood Cells | 77.51 [55.4 – 88.1] | **78.56** [67.5 – 87.4] | 75.91 [42.4 – 92.6] | 75.65 [40.7 – 91.3] |
| Wake | 61.63 [61.3 – 61.9] | 64.12 [63.3 – 64.8] | **65.88** [65.6 – 66.2] | 65.30 [65.0 – 65.6] |
| Light | 55.75 [55.4 – 56.0] | 56.70 [56.4 – 56.9] | **57.13** [56.8 – 57.4] | 56.65 [56.4 – 56.9] |
| Deep | 55.43 [54.7 – 56.1] | 55.21 [54.4 – 55.9] | **57.71** [56.9 – 58.3] | 57.46 [56.4 – 58.9] |
| Rem | 53.78 [53.3 – 54.2] | 54.78 [53.3 – 55.1] | **55.66** [55.2 – 56.0] | 54.45 [54.0 – 54.8] |
| **Regression - MAE (↓)** | **Chronos** | **Chronos-Bolt (210M)** | **MMR (7M)** | **MMR-Light (2M)** |
| Age | 8.95 [8.7 – 9.2] | 9.20 [8.9 – 9.5] | **8.37** [8.2 – 8.5] | 8.69 [8.5 – 8.8] |
| Sys. BP (Lab) | 11.15 [9.1 – 11.9] | **11.05** [8.9 – 11.8] | 11.28 [9.4 – 11.8] | 11.22 [9.5 – 11.8] |
| Dias. BP (Lab) | 7.83 [6.8 – 10.0] | 7.76 [6.8 – 9.9] | 7.76 [6.8 – 10.1] | **7.75** [6.6 – 10.1] |
| Sys. BP | 11.75 [11.6 – 11.8] | 11.69 [11.7 – 11.9] | **11.61** [11.5 – 11.7] | 11.63 [11.5 – 11.8] |
| Dias. BP | 9.47 [9.2 – 9.7] | 9.39 [9.1 – 9.6] | **9.32** [9.1 – 9.4] | 9.36 [9.1 – 9.6] |

### A.5.2 ADDITIONAL TIME SERIES MODEL EVALUATIONS

We evaluated CHRONOS-BOLT, the 210M-parameter successor to CHRONOS, on our downstream prediction tasks. As shown in Table 3, CHRONOS-BOLT consistently improves over the original CHRONOS on key demographic and cardiovascular classification tasks (e.g., higher AUROC on age, hypertension, and PVC detection) and achieves slightly lower MAE on most blood-pressure regression targets. Despite these improvements, our domain-pretrained MMR and MMR-LIGHT models achieve better performance on nearly all clinical endpoints, surpassing CHRONOS-BOLT by roughly 5% AUROC on age classification and 10% AUROC on PVC detection, and outperforming it on all but one regression target. These results suggest that domain-specific pretraining on large-scale wearable PPG helps in learning robust representations for various cardiovascular health tasks that general multivariate models may miss.

### A.6 EXPERIMENTAL SETTINGS

### A.6.1 DATA PREPROCESSING

Continuous streams of PPG data sampled at different rates are split into non-overlapping 10 s segments and each segment is then band-pass filtered using the zero-phase 0.5–10 Hz 4th-order Chebyshev filter Liang et al. (2018). All subsequent operations are applied to these filtered segments. Each 10 s segment is independently z-score normalized (per-segment standardization to zero mean and unit variance). Each segment is resampled to a common rate of 100 Hz using polyphase resampling, so every 10 s segment has length $T = 1000$. This grid ensures that, for a fixed wavelet configuration, each coefficient row corresponds to the same approximate frequency band across devices. Unless specified otherwise, we use per-band instance normalization: for each wavelet sub-band $c$ we subtract its mean over time and divide by its standard deviation over time within that segment.

*Wavelet coefficient map construction.* For MMR, segments at $100\ Hz$ are decomposed with a multi-level discrete wavelet transform (DWT) using PyWavelets Lee et al. (2019). The two highest-frequency detail bands lie outside the bandpass (cut at 10 Hz) and are therefore excluded from the 2D coefficient map construction. Because the DWT produces sub-bands at dyadic scales, coefficients at level $k$ have temporal length $T/2^k$. Each retained sub-band is upsampled to length $T$ using one-dimensional zero-order (nearest-neighbor) interpolation to form a dense, temporally aligned coefficient map of shape $(C \times T)$ (bands × time), suitable for patch-based masked auto encoding. In Appendix A.4 we show that replacing zero-order interpolation with linear or cubic interpolation yields very similar downstream performance.

### A.6.2 TRAINING SETUP

We pretrain MMR using the AdamW optimizer (Loshchilov & Hutter, 2017) with a base learning rate of $1 \times 10^{-4}$, a cosine decay schedule, and a linear warmup applied over the first 10% of $33K$ steps. Training is carried out with a batch size of 512, a weight decay of $1 \times 10^{-5}$, and gradient clipping at 1.0. To the PPG signal, we apply the same augmentations as LSM (Narayanswamy et al., 2024) prior to wavelet decomposition, specifically time-flipping, adding Gaussian noise, and stretching along the temporal axis. We apply instance-wise per band normalisation on the 2D cofficient map. The MMR model architecture follows the LSM-Small configuration (approximately 7M parameters), consisting of 8 encoder blocks with hidden size 256, 4 attention heads, and a feedforward size of 1024. For reconstruction during pretraining, we use a lightweight decoder composed of 2 blocks with hidden size 192 and 4 attention heads. The smaller variant, MMR-Light (approximately 2M parameters), follows the LSM-Tiny configuration from (Narayanswamy et al., 2024), with 4 encoder blocks (hidden size 192, 3 heads) and a lightweight decoder of 2 blocks (hidden size 128, 4 heads). Hyperparameter tuning was minimal and limited to a small grid search over candidate learning rates $\in \{10^{-2}, 10^{-3}, 10^{-4}\}$ and weight decay values $\in \{10^{-3}, 10^{-4}, 10^{-5}\}$. These sweeps and ablations were conducted on a subset of the pretraining dataset consisting of approximately 1 million data points. All experiments were performed on four Tesla T4 GPUs (16GB each) using distributed data parallel (DDP) training in PyTorch (Paszke et al., 2019).

### A.6.3 BASELINE METHODS

For the PAPAGEI family of models, we use the open-source pretrained weights released by Pillai et al. (2024) for PaPaGei-S, the morphology-aware pretraining model, which we refer to simply as PaPaGei. The model employs a ResNet-style convolutional encoder with 18 blocks, starting with 32 filters that double every four layers, and produces a 512-dimensional embedding through a projection head. The PAPAGEI-S variant additionally includes two mixture-of-expert heads for refining morphology-related indices (sVRI, IPA, SQI), resulting in approximately 5M parameters overall. Pretraining is performed on 57,000 hours of PPG data (around 20M segments) from large clinical datasets such as MIMIC-III (Johnson et al., 2016) and MESA (Chen et al., 2015), using a morphology-aware self-supervised objective with augmentations including cropping, Gaussian noise, flipping, negation, and magnitude scaling. In addition, we introduce a PAPAGEI-OURS baseline, which keeps the PAPAGEI-S architecture and morphology-aware objective fixed but retrains the model on the same smartwatch PPG pretraining cohort as MMR. This variant isolates the effect of MMR's multi-resolution inductive bias from differences in pretraining data.

For SimCLR (Chen et al., 2020), we adopt the same ResNet-18 backbone as PaPaGei ($\sim$5M parameters) and apply the standard NT-Xent loss (Sohn, 2016; Chen et al., 2020) with a temperature of $\tau = 0.2$. The augmentation pipeline includes random cropping (0.5), time flipping (0.2), negation (0.2), scaling (0.4), and Gaussian noise (0.35). For TF-C (Zhang et al., 2022a), we likewise use a ResNet-18 encoder, with a total of $\sim$10M parameters since two encoders are employed, and train with a time-frequency contrastive loss. Both SimCLR and TF-C are optimized using Adam with a base learning rate of $1 \times 10^{-4}$, a weight decay of $1 \times 10^{-5}$, a batch size of 128, and cosine learning rate scheduling. For the Masked Siamese Network (MSN) (Assran et al., 2022), we adopt transformer encoders with 4 attention heads and 12 layers. The latent dimension of the attention layers is 128, and the feedforward networks have a hidden size of 512, resulting in approximately 2.5M parameters overall. MSN is trained with a base learning rate of $5 \times 10^{-4}$, a weight decay of $1 \times 10^{-4}$, and linear warmup. Pretraining is performed on the same smartwatch PPG dataset as MMR, using a patch size of 10 and a masking ratio of 0.75.

As a lightweight baseline, we also implement the Statistical Features approach similar to (Pillai et al., 2024). Each 10s PPG segment is represented by a handcrafted feature vector: mean, standard deviation, 25th percentile, 50th percentile (median), 75th percentile, minimum, and maximum values (i.e., $[\text{mean}, \text{std}, p_{25}, p_{50}, p_{75}, \min, \max]$). These features are computed per segment and directly used as input to a random forest classifier or regressor, depending on the downstream task.

We include the open-source Chronos-T5 (Base, 200M parameters) (Ansari et al., 2024) as a large-scale time-series foundation model baseline. Chronos is based on the T5 encoder–decoder transformer architecture, adapted for time series by quantizing values into discrete tokens and applying mean-scaling normalization. It is trained autoregressively with cross-entropy loss on a large collection of public datasets and synthetic series, using additional augmentation strategies such as

TSMixup. Chronos represents a state-of-the-art zero-shot time-series model that is substantially larger than our other baselines, providing a complementary point of comparison for evaluating scale and domain adaptation effects. In addition, we also evaluate the Chronos-Bolt-Base configuration (∼210M parameters), a more recent variant of Chronos that incorporates architectural and training improvements for time-series modeling.

Finally, we include a masked time reconstruction (MTR) baseline designed to isolate the effect of MMR's multi-resolution objective. MTR uses exactly the same transformer backbone (8 encoder blocks with 256 hidden size, 4 attention heads and a decoder of 2 blocks with hidden dim 192 and 4 attention heads), pretraining data and optimization hyperparameters as MMR, but is trained to reconstruct masked segments of the time-domain PPG signal rather than multiscale wavelet coefficients.

### A.6.4 EVALUATION SETTINGS

For all probing tasks, we train random forest classifiers and regression models using a *grouped, stratified 5-fold cross-validation* scheme. Grouping is performed at the subject level so that all segments from a given user are assigned to a single fold, fully preventing subject-level data leakage between training and test sets. Stratification preserves the marginal distribution of target classes across folds, which is important for our imbalanced outcomes. Final numbers are mean scores obtained by averaging the per-fold values across the five folds; the "min" and "max" entries in the tables ( 8, 9) correspond to the minimum and maximum fold-level scores. Labels are computed at the segment level. For sleep stage classification tasks, we report scores and the boostrapped confidence intervals on a single fold given the large number of segments. To further avoid any leakage between pretraining and evaluation, we enforce strict user-wise splits: no user present in the pretraining data is used in any probing experiment.

### A.7 DATASET DETAILS

We pretrain on data collected from various types of REDACTED smartwatches. The cohort spans multiple smartwatch versions (device versions v1–v8; see Fig. 8), with substantial variation in user counts across devices; the group labeled "v4" corresponds to users for whom the exact device version is unknown (NA). These datasets provide diverse signals collected from diverse user groups, closely reflecting real-world conditions and making them representative for PPG-based wearable applications. Across pretraining and test splits, we ensure user-wise separation, but device versions are shared rather than being strictly exclusive to a single split.

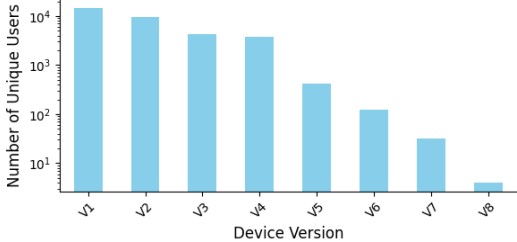

Figure 8: Device vs User Count for our Pretraining Cohort

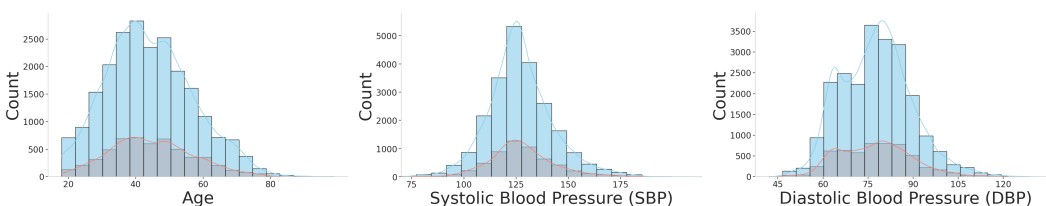

Figure 9: Hypertension Free Living Data Statistics

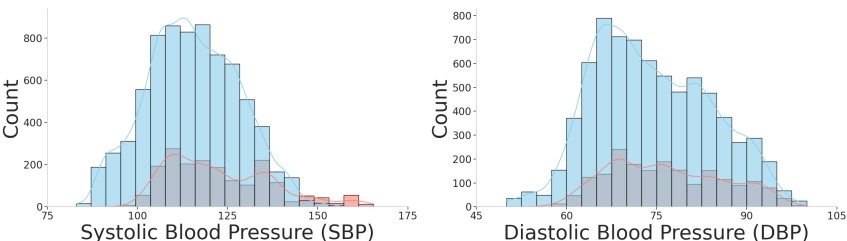

Figure 10: Hypertension Lab Data Statistics

Table 4: Downstream datasets. We report the number of users and the count of positive and negative segments across the entire dataset.

| Task | Setting | Users | Positive | Negative |
|---|---|---|---|---|
| Hypertension | Lab (protocol) | 63 | 646 | 3,595 |
| | Naturalistic (field) | ~2K | 5,952 | 8,189 |
| PVC Detection | Naturalistic (field) | 137 | 40,780 | 442,484 |
| *Laboratory Tests* | | | | |
| Sodium | Clinical reports | 45 | 9,714 | 8,721 |
| Potassium | Clinical reports | 45 | 10,103 | 9,509 |
| Creatinine | Clinical reports | 39 | 8,176 | 8,879 |
| Carbon Dioxide | Clinical reports | 37 | 7,846 | 9,141 |
| Hemoglobin | Clinical reports | 29 | 6,429 | 6,286 |
| White Blood Cells | Clinical reports | 28 | 7,384 | 6,380 |
| Age Classification | Demographic | ~15K | 13,853 | 31,826 |

### A.7.1 DOWNSTREAM DATASETS

**User cohort size.** We evaluate our representations on a diverse set of downstream tasks spanning controlled laboratory cohorts, large-scale free-living data, and public benchmarks:

1. **Hypertension–Lab:** 63 users, collected in a laboratory data-collection setting.

2. **Hypertension–Free Living:** approximately 15,000 users collected in free-living conditions, providing a large and heterogeneous evaluation cohort.

3. **PVC Detection:** 137 users with a large number of labeled segments (~400,000).

4. **Laboratory Biomarkers:** Collected from the same cohort, but the number of users ($\approx$28–45 users) across tasks – sodium, potassium, creatinine, carbon dioxide, hemoglobin, and white blood cells– varies.

5. **Age:** age classification and regression are defined on the same cohort as the Hypertension–Free Living dataset.

6. **Sleep Stage Classification:** we use the Dreamt dataset Wang et al. (2024a;b) hosted on PhysioNet Goldberger et al. (2000), comprising 100 users.

### A.7.2 TASKS

**Hypertension classification.** We define hypertension as a binary classification task based on clinical guidelines. Individuals are labeled as *Hypertensive* (label 1) if their systolic blood pressure is $\geq 130$ mmHg or diastolic blood pressure is $\geq 80$ mmHg, and *Normal* (label 0) otherwise. To reduce label uncertainty near the diagnostic cutoffs, we apply buffer thresholds of $\pm 8$ mmHg around these boundaries and exclude measurements within the buffer from training.

**PVC detection.** Premature Ventricular Contractions (PVCs) are early heartbeats originating in the ventricles (Cha et al., 2012) and may indicate underlying cardiac conditions or increased arrhythmia

Table 5: Sleep staging dataset (Dreamt) summary.

| Class | Label | #Segments (10 s) |
|---|---|---|
| Wake | 0 | 56,127 |
| Light (N1+N2) | 1 | 146,085 |
| Deep (N3) | 2 | 8,112 |
| REM | 3 | 25,095 |

risk. Prior studies have investigated PVC detection using PPG data (Han et al., 2020; Solosenko & Marozas, 2014). In our setting, we define a binary task at the user level: high PVC burden (label 1) versus low PVC burden (label 0).

**Laboratory tests.** For each laboratory biomarker, we formulate a binary classification task where abnormal values are labeled as class 1 and values within the reference range as class 0, following clinical descriptions in (National Library of Medicine (US), 2020):

- **Sodium:** Elevated sodium (hypernatremia) is linked to dehydration or adrenal gland/kidney dysfunction.
- **Potassium:** High potassium (hyperkalemia) may cause cardiac arrhythmias; low potassium (hypokalemia) is associated with muscle weakness, fatigue, and rhythm disturbances.
- **Creatinine:** Elevated creatinine reflects impaired kidney function and may indicate acute kidney injury, chronic kidney disease, or other kidney problems.
- **Carbon dioxide:** Abnormal carbon dioxide levels suggest an acid–base imbalance. Low levels may indicate metabolic acidosis or respiratory alkalosis, whereas high levels may reflect metabolic alkalosis or chronic respiratory acidosis.
- **Hemoglobin:** Low hemoglobin levels may indicate anemia, which can result from iron deficiency, chronic disease, or blood loss.
- **White blood cells (WBC):** Elevated WBC (leukocytosis) may be seen with infection, inflammation, stress, or blood disorders. Low WBC (leukopenia) can indicate bone marrow suppression, viral infection, or autoimmune conditions.

**Age prediction.** We consider both classification and regression tasks on the Hypertension–Free Living cohort. For classification, we define a binary label with *older* adults (age $> 50$ years) as class 1 and *younger* adults (age $\leq 50$ years) as class 0. For regression, the target is chronological age in years.

**Sleep stage classification (Dreamt).** For sleep staging, we use the Dreamt dataset Wang et al. (2024a) hosted on PhysioNet Goldberger et al. (2000). We perform 4-class classification at the 10 s segment level, merging N1 and N2 into a single *Light* sleep class. The resulting classes are Wake, Light (N1+N2), Deep (N3), and REM. We report one-vs-rest AUROC for each class.

## A.8 EXAMPLE VISUALIZATION OF DISCRETE WAVELET TRANSFORMS OF WEARABLE PPG SIGNAL

In this section, we present visualizations illustrating the transformation of PPG signals into DWT coefficients (Fig. 11) and their subsequent processing within our masked modeling framework. The patchified coefficient maps are heavily masked, and the MMR encoder–decoder is trained to reconstruct the missing subbands (Fig. 12). These examples demonstrate how the DWT captures multi-resolution structure and how MMR exploits this representation to recover meaningful time–frequency information from incomplete inputs.

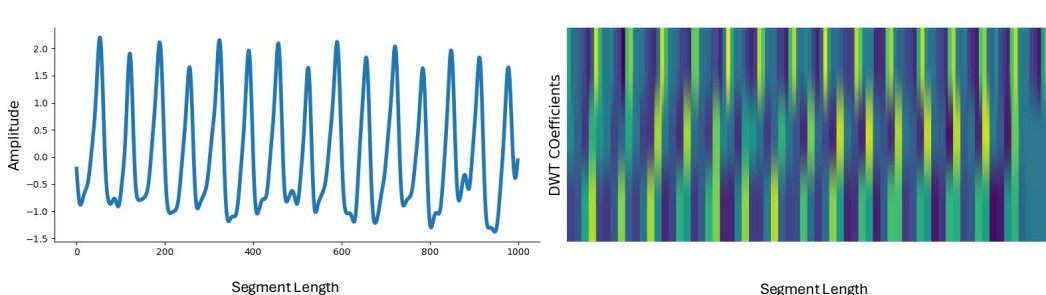

Figure 11: Example PPG signal (left) and its corresponding 2-D representation of discrete wavelet transform (DWT) coefficients (right). The DWT decomposes the signal into multi-resolution sub-bands, where higher-frequency detail coefficients appear at the top and the low-frequency approximation band at the bottom, providing a time–frequency representation.

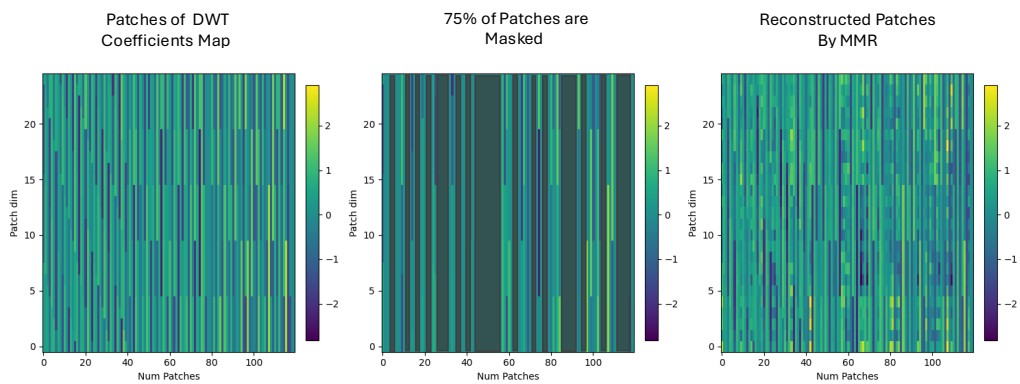

Figure 12: Illustration of the masked modeling framework applied to DWT coefficients. Left: original patchified DWT coefficient map of the PPG signal. Middle: masked input where 75% of patches are randomly removed. Right: reconstructed patches generated by the MMR model, demonstrating its ability to recover missing time–frequency structure from partial observations.

## A.9 ADDITIONAL CLASSIFICATION RESULTS FOR MMR AND BEST PERFORMING BASELINES

Table 6 and Table 7 report the full set of AUROC scores for model scaling and data scaling experiments. These results show that larger models (27M parameters) achieve the highest performance on several endpoints, including Hypertension-Lab and Sodium prediction, while the mid-sized model remains competitive across most tasks. Similarly, increasing pretraining data size from 1M to 17M segments yields consistent gains for hypertension and PVC detection, with more modest improvements for certain lab biomarkers such as WBC and creatinine. We note that these results are obtained using a level-4 decomposition of the pretraining dataset and may therefore differ from the main results. Here, our goal is only to assess whether model and data scaling effects are present in the pretraining task; and the same qualitative behavior is expected to hold for a level-3 decomposition.

| Classification AUROC (↑) | 2 Million | 7 Million | 27 Million |
|---|---|---|---|
| Hypertension-Lab | 71.50 | 71.18 | **80.54** |
| Hypertension | 67.08 | **68.12** | 66.65 |
| PVC Detection | 81.30 | **82.04** | 81.41 |
| *Laboratory Tests* | | | |
| Carbon Dioxide | **63.95** | 63.10 | 62.57 |
| Creatinine | 57.28 | 54.82 | **58.26** |
| Hemoglobin | 53.19 | 52.63 | **54.57** |
| Potassium | 73.23 | **74.27** | 71.34 |
| Sodium | 57.08 | 60.54 | **64.45** |
| White Blood Cells | 75.53 | 75.71 | **76.12** |

Table 6: **MMR Model Scaling Results:** This table reports *mean AUROC* on the 5-fold cross-validation performance metrics for MMR models of increasing scale (2 million, 7 million, and 27 million parameters). Each column represents a model size, while rows correspond to the predictive tasks (e.g., hypertension, PVC, WBC).

| Classification AUROC (↑) | 1 Million | 5 Million | 17 Million |
|---|---|---|---|
| Hypertension-Lab | 67.99 | 69.06 | **71.18** |
| Hypertension | 66.08 | 67.50 | **68.12** |
| PVC | 76.62 | 79.84 | **82.04** |
| *Laboratory Tests* | | | |
| Carbon Dioxide | **63.80** | 63.54 | 63.10 |
| Creatinine | **59.60** | 54.60 | 54.82 |
| Hemoglobin | 52.21 | 53.89 | 52.63 |
| Potassium | 74.03 | 74.92 | 74.27 |
| Sodium | 57.69 | 57.25 | **60.54** |
| White Blood Cells | **76.80** | 75.16 | 75.71 |

Table 7: **MMR Data Scaling Results:** This table reports *mean AUROC* on the 5-fold cross-validation performance metrics for MMR model with increasing pretraining data scale (1 million, 5 million, and 17 million segments). Each column represents pretraining data size, while rows correspond to the predictive tasks (e.g., hypertension, PVC, WBC).

## A.10 ANALYZING LEARNED REPRESENTATIONS FOR DOWNSTREAM TASKS

We present t-SNE visualizations of patient-level embeddings learned by MMR and PaPaGei for the Hypertension task (Free Living). Figure 13 (left) shows the embeddings colored by age bins. While the different age groups are not perfectly separable, the latent space reveals slight clustering, with patients above 50 years more prominently shifting toward the left side of the embedding. This indicates that the model encodes some age-related demographic information. We found that, beyond patient-level discriminability, the learned embeddings capture clear physiological structure. As shown in Figure 13, t-SNE visualizations of participant-level mean embeddings colored by heart rate reveal a smooth gradient from low to high values. This indicates that the MMR model encodes physiologically meaningful variation rather than random alignment. Such clustering is not observed in t-SNE of PaPaGei embeddings in Fig. 14.

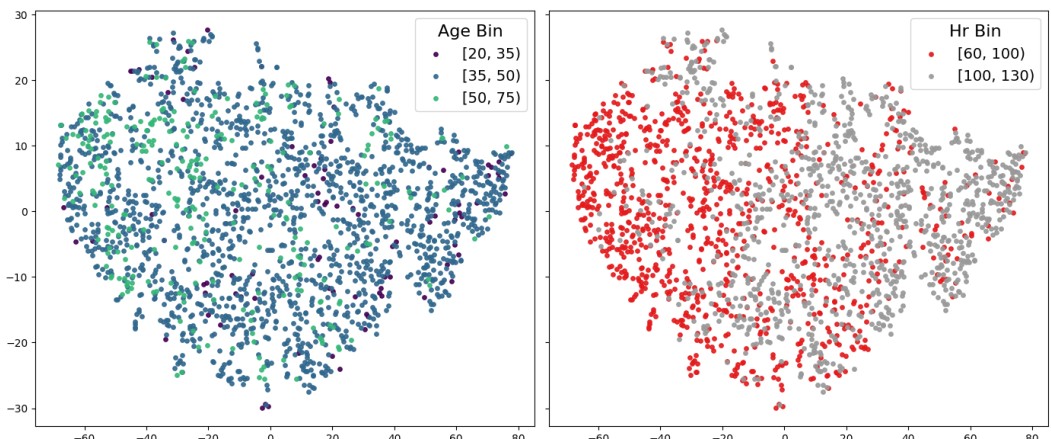

Figure 13: t-SNE visualization of patient-level embeddings (mean of all segments per patient) learned by MMR for the Hypertension Task – Free Living. Left: embeddings colored by age bins show slight clustering, consistent with prior observations (Narayanswamy et al., 2024). Right: embeddings colored by heart rate bins reveal a gradient, indicating that the representations capture meaningful physiological variability.

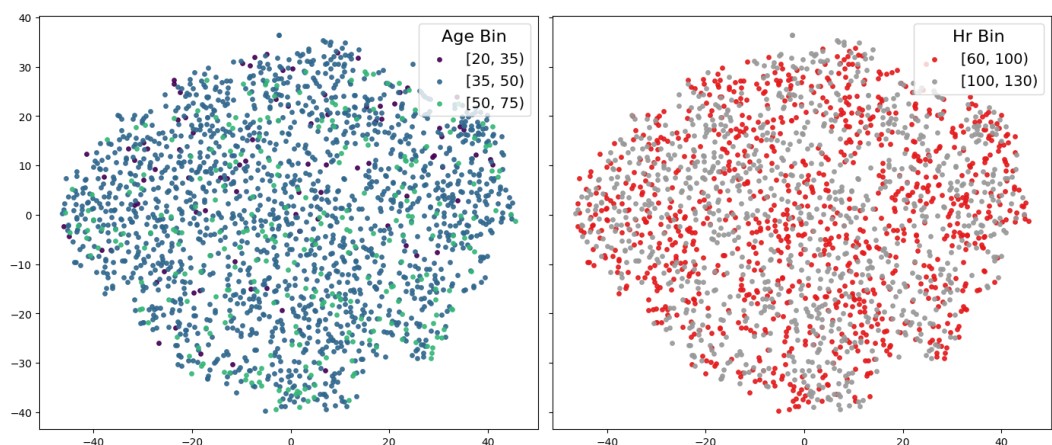

Figure 14: t-SNE visualization of PaPaGei embeddings for Hypertension Task – Free Living.

## A.11 ADDITIONAL METRICS: F1-SCORE

We report F1-scores after applying class reweighting to provide a more balanced comparison across rare classes. Both MMR and MMR-Light achieve strong F1-scores across tasks. In the self-supervised comparison (Table 8), MMR/MMR-Light obtain the best score on 9 out of 14 tasks, while the remaining wins are distributed among SimCLR, MSN, and MTR. In the open-source comparison (Table 9), MMR/MMR-Light perform even more prominently, achieving the top score on 11 out of 14 tasks, with only a small number of tasks favoring Chronos or PaPaGei.

Table 8: Comparison with Self-Supervised Baselines–F1-score. Best downstream evaluation scores are shown in **bold**.

| Classification - F1-Score (↑) | SimCLR (5M) | MSN (2.5M) | TF-C (10M) | MTR (7M) | MMR (7M) | MMR-Light (2M) |
|---|---|---|---|---|---|---|
| Age | 49.47 [48.2 – 50.6] | 53.35 [52.1 – 55.0] | 51.78 [50.4 – 53.0] | 56.50 [54.8 – 58.2] | **58.91** [57.0 – 61.1] | 56.68 [54.9 – 58.7] |
| Hypertension - Lab | 20.95 [9.6 – 27.1] | 14.72 [4.8 – 23.2] | 26.27 [6.1 – 42.3] | 36.62 [15.7 – 58.3] | 32.95 [11.5 – 61.4] | **37.63** [13.2 – 68.0] |
| Hypertension - Free Living | 52.80 [51.5 – 53.8] | 54.80 [53.8 – 55.8] | 52.40 [49.9 – 54.2] | 58.97 [57.6 – 60.0] | **59.64** [58.7 – 61.7] | 58.51 [57.6 – 59.9] |
| PVC | 26.09 [13.3 – 38.3] | 32.26 [19.0 – 42.6] | 21.30 [10.8 – 28.7] | 37.96 [18.9 – 52.4] | **39.22** [18.9 – 56.0] | 38.16 [16.2 – 50.3] |
| Carbon Dioxide | 51.88 [23.1 – 63.1] | 50.26 [31.5 – 62.7] | 50.92 [36.1 – 71.0] | **52.68** [33.5 – 70.5] | 45.04 [30.5 – 64.6] | 42.87 [25.3 – 59.6] |
| Creatinine | 46.57 [14.2 – 67.9] | **51.40** [22.7 – 68.5] | 46.13 [20.3 – 65.9] | 43.92 [34.2 – 64.7] | 46.15 [36.3 – 61.8] | 47.31[35.4 – 62.4] |
| Hemoglobin | 49.55 [25.0 – 71.8] | 48.54 [23.8 – 74.8] | 37.43 [16.9 – 57.8] | 47.03 [27.0 – 77.5] | **53.30** [22.1 – 74.9] | 52.63 [21.0 – 76.8] |
| Potassium | 66.77 [57.7 – 69.5] | 67.85 [53.9 – 74.0] | 68.29 [61.2 – 72.8] | **69.36** [57.3 – 75.1] | 65.11 [52.6 – 75.2] | 65.27 [55.7 – 71.7] |
| Sodium | 50.09 [37.6 – 59.4] | 51.73 [34.7 – 61.5] | 47.72 [27.3 – 59.8] | 49.74 [27.4 – 58.0] | 52.77[38.5 – 60.3] | **52.78** [31.1 – 62.0] |
| White Blood Cells | 67.60 [54.3 – 79.9] | 66.25 [45.2 – 79.4] | 67.52 [53.8 – 77.6] | **68.02** [49.2 – 82.3] | 61.30 [49.2 – 80.3] | 65.94 [50.7 – 79.0] |
| Wake | 41.73 [41.3 – 42.1] | 42.93 [42.6 – 43.3] | 43.63 [43.3 – 43.9] | 43.58 [43.3 – 43.9] | **44.01** [43.7 – 44.4] | 43.84 [43.5 – 44.2] |
| Light | **55.97** [55.7 – 56.2] | 50.56 [50.3 – 50.8] | 44.47 [44.2 – 44.7] | 49.44 [49.2 – 49.7] | 51.02 [50.7 – 51.3] | 50.95 [50.7 – 51.2] |
| Deep | 11.60 [10.9 – 12.2] | 10.54 [9.96 – 11.13] | 11.04 [10.5 – 11.6] | 12.83 [12.1 – 13.5] | 10.91 [10.2 – 11.6] | **13.62** [12.8 – 14.4] |
| Rem | 14.96 [14.5 – 15.3] | 13.42 [13.0 – 13.8] | 15.73 [15.3 – 16.2] | 13.74 [13.4 – 14.2] | 14.40 [14.0 – 14.8] | **15.61** [15.2 – 16.0] |

Table 9: Comparison with Open-source Pretrained Models and Statistical Features–F1-score . Best downstream evaluation scores are shown in **bold**.

| Classification - F1-score (↑) | Stat. Feat. | Chronos (200M) | PaPaGei (5M) | PaPaGei-Ours (5M) | MMR (7M) | MMR-Light (2M) |
|---|---|---|---|---|---|---|
| Age | 53.25 [52.4 – 54.2] | 49.77 [48.9 – 50.9] | 45.06 [44.4 – 45.8] | 49.44 [48.2 – 50.9] | **58.91** [57.0 – 61.1] | 56.68 [54.9 – 58.7] |
| Hypertension - Lab | 30.33 [18.2 – 52.7] | 14.93 [2.2 – 25.0] | 17.16 [9.9 – 27.4] | 21.29 [11.1 – 39.0] | 32.95 [11.5 – 61.4] | **37.63** [13.2 – 68.0] |
| Hypertension - Free Living | 51.27 [49.2 – 54.4] | 45.50 [43.2 – 46.7] | 47.20 [46.1 – 48.0] | 45.24 [43.7 – 47.7] | **59.64** [58.7 – 61.7] | 58.51 [57.6 – 59.9] |
| PVC | 23.58 [11.3 – 33.8] | 24.33 [12.3 – 34.9] | 26.75 [13.7 – 40.6] | 28.31 [15.4 – 37.5] | **39.22** [18.9 – 56.0] | 38.16 [16.2 – 50.3] |
| Carbon Dioxide | 41.95 [22.8 – 54.9] | 50.06 [32.9 – 59.6] | **46.81** [22.8 – 62.1] | 45.14 [26.4 – 54.8] | 45.04 [30.5 – 64.6] | 42.87 [25.3 – 59.6] |
| Creatinine | 39.76 [17.3 – 56.5] | 41.84 [30.8 – 57.0] | 42.05 [17.4 – 56.8] | 41.53 [30.8 – 54.4] | 46.15 [36.3 – 61.8] | **47.31** [35.4 – 62.4] |
| Hemoglobin | 35.55 [25.4 – 49.3] | 37.21 [19.8 – 73.8] | 38.12 [28.7 – 49.8] | 37.00 [22.7 – 45.1] | **53.30** [22.1 – 74.9] | 52.63 [21.0 – 76.8] |
| Potassium | 58.50 [49.7 – 64.7] | 65.71 [63.6 – 68.4] | 62.57 [57.6 – 65.5] | 60.99 [56.6 – 64.2] | 65.11 [52.6 – 75.2] | **65.27** [55.7 – 71.7] |
| Sodium | 47.68 [29.2 – 63.6] | 54.32 [40.4 – 67.1] | 48.21 [36.2 – 60.5] | 45.99 [34.3 – 58.0] | 52.77 [38.5 – 60.3] | **52.78** [31.1 – 62.0] |
| White Blood Cells | 59.12 [39.4 – 71.3] | 65.99 [57.5 – 75.1] | 59.85 [46.9 – 69.0] | 62.67 [43.1 – 72.1] | 61.30 [49.2 – 80.3] | **65.94** [50.7 – 79.0] |
| Wake | 40.80 [40.38 – 41.18] | 40.26 [39.90 – 40.63] | 41.74 [41.37 – 42.15] | 40.53 [40.13 – 40.94] | **44.01** [43.70 – 44.39] | 43.84 [43.52 – 44.17] |
| Light | 51.70 [51.45 – 51.97] | 54.17 [53.91 – 54.42] | 45.85 [45.57 – 46.14] | **55.49** [55.22 – 55.77] | 51.02 [50.76 – 51.31] | 50.95 [50.68 – 51.22] |
| Deep | 9.31 [8.86 – 9.74] | 8.93 [8.22 – 9.63] | 10.72 [9.98 – 11.29] | 10.01 [9.40 – 10.55] | 10.91 [10.16 – 11.65] | **13.62** [12.88 – 14.43] |
| Rem | 15.47 [15.11 – 15.85] | 15.47 [15.11 – 15.85] | **17.28** [16.96 – 17.60] | 15.61 [15.23 – 16.02] | 14.40 [14.01 – 14.83] | 13.92 [13.55 – 14.31] |

