# OpenReview forum: "Wavelet-Driven Masked Multiscale Reconstruction for PPG Foundation Models"
_ICLR.cc/2026/Conference — Submitted to ICLR 2026_

### Official Review · Reviewer_H7f6 · 2025-10-30

**Soundness:** 3
**Presentation:** 3
**Contribution:** 2
**Rating:** 4
**Confidence:** 4

**Summary:**

This paper proposes Masked Multiscale Reconstruction, a self-supervised pretraining framework for wearable PPG signals. The key idea is to transform each 10-s segment into multi-resolution wavelet coefficients, randomly mask 75% of small temporal patches across sub-bands, and train a ViT encoder–decoder to reconstruct the missing coefficients. The authors pretrain on ~17M PPG segments from ~32K smartwatch users, then evaluate frozen embeddings with simple downstream heads across 13 health-related tasks. They report consistent gains or parity versus strong baselines and competitive/open-source models, with notable improvements on free-living hypertension and PVC detection; they also include data/model scaling and ablations.

**Strengths:**

1.	On free‑living hypertension, MMR achieves 68.1 AUROC versus 62.9 for TF‑C and 60.9 for SimCLR; for PVC detection, MMR reaches 82.0 AUROC versus 70–74 for baselines. Across classification tasks, MMR also outperforms PaPaGei and Chronos on average.
2.	Treating PPG in the time–frequency domain with wavelets matches physiological non‑stationarity and multi‑scale rhythms, and masked reconstruction across sub‑bands encourages cross‑scale feature sharing.
3.	Ablations and scaling analyses offer practical guidance: Haar wavelets often perform best; moderate decomposition depth (L3–L4) helps hypertension/PVC; smaller patches (1×25) preserve fine events; and increasing data volume improves key tasks.

**Weaknesses:**

1.	The core idea is masked autoencoding in a time–frequency domain with a fixed DWT tokenizer. Closely related frequency-aware MAE variants already exist, and the contribution feels incremental.
2.	Using a predefined DWT front-end may underfit device-specific characteristics and miss discriminative sub-band boundaries. It will be better to explore the learnable filterbanks or jointly trained spectral tokenizers.
3.	Converting multi-rate sub-bands into a 2D coefficient map via interpolation risks aliasing and amplitude distortion. The interpolation kernel, anti-aliasing, and per-band normalization choices are not justified or stress-tested.
4.	Uniform random masking likely biases learning toward low-frequency shortcuts. Missing band-aware, curriculum, event-centric, or rarity-aware masking that would force recovery of high-frequency/rare cues (e.g., ectopy).
5.	No clear rebalancing (e.g., class weights, focal loss), threshold optimization, or cost-sensitive analysis; this can suppress performance on rare events.
6.	Tables 1–2 show that MMR is not SOTA on several classification endpoints and is sometimes matched or outperformed by MMR-Light or baselines. The paper does not explain these gaps or provide a failure-mode, leaving the causes of underperformance unclear.

**Questions:**

1.	Could you clarify how your method is substantively different from existing frequency-aware MAE variants and why fixed-DWT masked autoencoding constitutes a novel contribution?
2.	Will you evaluate learnable filterbanks or jointly trained spectral tokenizers to address potential underfitting to device-specific characteristics and missed discriminative sub-band boundaries?
3.	How were the interpolation kernel, anti-aliasing, and per-band normalization chosen when converting multi-rate sub-bands into a 2D map, and can you provide stress tests showing minimal aliasing/amplitude distortion?
4.	Can you justify uniform random masking and report results for band-aware, event-centric/curriculum, or rarity-aware masking that better targets high-frequency/rare cues?
5.	Did you apply class rebalancing (class weights/focal loss), threshold optimization, or cost-sensitive analysis? If not, can you add these and report their impact on rare-event metrics?
6.	In Table 1&2, MMR is not SOTA on several classification endpoints and in a few cases is even matched or exceeded by MMR-Light or baselines (SimCLR/TF-C). Could you explain these gaps and provide a brief failure-mode analysis?

---

> ### Author Response · Authors · 2025-11-24
> **Rebuttal**
>
> ### Weakness/Question 1: Comparison with Frequency-based MAE Methods
> > The core idea is masked autoencoding in a time–frequency domain with a fixed DWT tokenizer. Closely related frequency-aware MAE variants already exist, and the contribution feels incremental.
> Could you clarify how your method is substantively different from existing frequency-aware MAE variants and why fixed-DWT masked autoencoding constitutes a novel contribution?
>
> We thank the reviewer for this thoughtful comment. We acknowledge that masked autoencoding itself is well established and is not the novelty of our work. Our contribution lies in how we instantiate masked reconstruction for PPG via a wavelet-based, multi-resolution time–frequency representation, and we show that this design materially changes performance. We discuss comparison to prior approaches in the following.
>
> **Comparison to frequency-aware MAE variants.** FreqMAE [1] uses a Short-Time Fourier Transform (STFT) for time-frequency representation of signals from different modalities. BioFame [2] employs only a fixed-window Discrete Fourier Transform (DFT) for modelling the raw signal. Instead, we propose to represent input signals using a wavelet-based DWT hierarchy in combination with a cross-resolution reconstruction objective. Prior approaches relying on fixed-window STFT/DFT decompositions suffer from inherent limitations due to their **uniform time-frequency resolution**: a single fixed analysis window constrains all frequency components to share the same temporal support, which leads to suboptimal modelling of short-lived high-frequency transients and coarse characterization of long-range low-frequency structure. This rigid resolution also amplifies the impact of the Fourier uncertainty trade-off [3], preventing these representations from adapting to the nonstationary, multi-scale nature of many real-world signals. In contrast, the multiresolution properties of the DWT provide **scale-dependent localization**, adaptively allocating fine temporal resolution to high-frequency content and fine spectral resolution to low-frequency content, yielding richer and more discriminative representations for reconstruction-based pretraining. We also note that DWT-based signal representations have been used effectively for biosignals in prior work, but not in combination with a masked reconstruction objective [4,5].
>
> **Comparison to TF-C.** TF-C uses a contrastive learning formulation as the self-supervised pretraining objective but MMR leverages masked reconstruction instead. In Table 1, we add TF-C as a competitive frequency-aware baseline and find that MMR outperforms TF-C on 17 of 19 downstream tasks, indicating that the proposed multiscale wavelet reconstruction objective yields more transferable representations than existing fixed-resolution time–frequency consistency objectives.
>
>
> **Comparison to Time-Domain MAE.** To further isolate the effect of the DWT front end from generic MAE, we add a time-domain masked autoencoder baseline (Masked Time Reconstruction (MTR) in Table 1) that shares the same ViT encoder–decoder, masking ratio, optimizer, and schedule but reconstructs patchified raw PPG signals. In Table 1 (revised), we find that MMR outperforms MTR on 17 of 19 downstream tasks, showing that changing the raw signal representation to a Discrete Wavelet front end and reconstruction target leads to meaningful gains beyond standard time-domain MAE.
>
> [1] FreqMAE: Frequency-Aware Masked Autoencoder for Multi-Modal IoT Sensing, Proceedings of the ACM Web Conference 2024
>
> [2] Frequency-Aware Masked Autoencoders for Multimodal Pretraining on Biosignals, ICLR 2024 Learning from Time Series for Health Workshop
>
> [3] Oppenheim, A. V. and Schafer, R. W. Discrete-Time Signal Processing (Chap. 10). Pearson, 2010
>
> [4] A comprehensive review of techniques for processing and analyzing fetal heart rate signals. Sensors, 21 (18):6136, 2021
>
> [5] Analysis of Fetal Heart Rate in Healthy and Pathological Pregnancies Using Wavelet-Based Features. In Proceedings of the 2007 29th Annual International Conference of the IEEE Engineering in Medicine and Biology Society

---

> > ### Author Response · Authors · 2025-11-24
> > **Rebuttal**
> >
> > ### Weakness/Question 2: Learnable Filterbanks
> > > Using a predefined DWT front-end may underfit device-specific characteristics and miss discriminative sub-band boundaries. It will be better to explore the learnable filterbanks or jointly trained spectral tokenizers.
> > Will you evaluate learnable filterbanks or jointly trained spectral tokenizers to address potential underfitting to device-specific characteristics and missed discriminative sub-band boundaries?
> >
> > We appreciate the reviewer's comment. Our goal in this work is to explicitly inject a strong discrete **wavelet–based inductive bias** into large-scale PPG pretraining, so we intentionally adopt a predefined DWT front-end rather than a fully learnable filterbank. Discrete wavelet transforms have been in many PPG/Biosignalsworks for many  based used in downstream tasks such as  cardiovascular health tasks and arrhythmia detection [1] [2], which motivates our choice of a wavelet-based front-end. In this work, we therefore focus on a fixed DWT front-end: this design stabilizes training, reduces the number of learnable parameters, and allows us to clearly isolate the contribution of the wavelet inductive bias, without conflating it with the optimization of a learned front-end. Across a diverse suite of tasks (Tab. 1 and Tab. 2), we observe that the inductive bias helps in improving performance across competitive  baselines. We consider building learnable filterbanks or jointly trained spectral tokenizers as promising directions for future work in advancing wavelet-based foundation models for PPG and other physiological biosignals.
> >
> > [1]Merino-Monge, Manuel, et al. "Heartbeat detector from ECG and PPG signals based on wavelet transform and upper envelopes." Physical and Engineering Sciences in Medicine 46.2 (2023): 597-608.
> >
> > [2]Yang, Chengming, et al. "Using PPG signals and wearable devices for atrial fibrillation screening." IEEE Transactions on Industrial Electronics 66.11 (2019): 8832-8842.
> >
> >
> > ### Weakness/Question 3: Stress-Testing Interpolation Kernels and Normalization
> > > Converting multi-rate sub-bands into a 2D coefficient map via interpolation risks aliasing and amplitude distortion. The interpolation kernel, anti-aliasing, and per-band normalization choices are not justified or stress-tested.
> > How were the interpolation kernel, anti-aliasing, and per-band normalization chosen when converting multi-rate sub-bands into a 2D map, and can you provide stress tests showing minimal aliasing/amplitude distortion?
> >
> > In the revised manuscript, we have added details on our  choices constructing 2D coefficients (See App. A.6.1) and added ablations to stress test our design choices (see App. A.4). We discuss them briefly as follows.
> >
> > **Stress tests.** In App. A.4 and Fig. 6 in updated manuscript, we additionally report two stress tests:
> >
> > **(i) Interpolation kernel ablation:** Replacing zero-order interpolation with linear or cubic interpolation changes the mean downstream AUROC on representative tasks  by at most ≈0.2, with error bars overlapping zero, indicating no systematic gain from alternative kernels and low sensitivity to this choice.
> >
> > **(ii) Normalization ablation:** comparing per-band instance normalization vs. global instance normalization over the full 2D map yields mean performance differences close to zero (Fig 6), suggesting that our specific normalization scheme is not a key driver of results.
> >
> >
> > ### Weakness/Question 4: Random Masking Justification
> > > Uniform random masking likely biases learning toward low-frequency shortcuts. Missing band-aware, curriculum, event-centric, or rarity-aware masking that would force recovery of high-frequency/rare cues (e.g., ectopy).
> > Can you justify uniform random masking and report results for band-aware, event-centric/curriculum, or rarity-aware masking that better targets high-frequency/rare cues?
> >
> > Our default masking is uniform over tokens in the band×time grid. In the original submission we already compared three strategies: row-wise (band-aware), cross-scale (column-wise), and uniform random and found that uniform random achieved the best average AUROC across tasks (see Fig. 7.(a)), which is why we adopted it as the default. This is also consistent with standard MAE traning practice and is used in our time-domain MAE baseline (MTR; see Table 1).
> >
> > In response to the reviewer's comment, we additionally introduce a frequency-guided masking variant in App. A.4: we compute the FFT magnitude for each patch, use the frequency-domain energy as a score, and mask high-energy patches. As shown in Fig. 7.(a), frequency-guided masking slightly improves some tasks (WBC, Sodium) but degrades others (e.g., BP-Lab, BP, Hemoglobin, Potassium), leading to no consistent advantage over uniform random masking when averaged across tasks.

---

> > > ### Author Response · Authors · 2025-11-25
> > > **Rebuttal**
> > >
> > > ### Weakness6/ Question 6: Failure Mode Analysis
> > > > In Table 1&2, MMR is not SOTA on several classification endpoints and in a few cases is even matched or exceeded by MMR-Light or baselines (SimCLR/TF-C). Could you explain these gaps and provide a brief failure-mode analysis?”
> > > We thank the reviewer for this observation. We provide detailed clarification in the following.
> > >
> > > **MMR performs better on many tasks.** Tables 1–2 evaluate a broad suite of classification  and regression tasks spanning cardiovascular health and sleep staging. Across this heterogeneous set, MMR and its lightweight variant MMR-Light generally outperform prior methods. The largest gains are observed on tasks such as PVC detection, hypertension classification, and binarized age prediction. All baselines (SimCLR, MSN, TF-C, MTR, PaPaGe-Ours) are trained with the same pretraining data and protocol, making the comparison strict and controlled.
> > >
> > > **Where MMR fails.** For the lab endpoints MMR does not clearly win (e.g. Hemoglobin, WBC), however, performance across methods is varied, underscoring the challenging nature of mapping PPG signals to lab-report outcomes. Additionally, MMR like other foundation models is pretrained with a single SSL objective to learn general representations, so a frozen probe may not be optimal for all the 19 heterogeneous tasks. The underperforming cases likely reflect objective endpoint mismatch (wavelet SSL emphasizes waveform changes/global rhythms in a 10 sec PPG segment, patterns that transfer well to rhythm/hypertensions tasks, but less to indirect labels like  ubnormal lab-value classification).

---

> > > > ### Author Response · Authors · 2025-12-01
> > > > **Rebuttal**
> > > >
> > > > ### Weakness5/Question 5 Class Rebalancing
> > > > > No clear rebalancing (e.g., class weights, focal loss), threshold optimization, or cost-sensitive analysis; this can suppress performance on rare events. Did you apply class rebalancing (class weights/focal loss), threshold optimization, or cost-sensitive analysis? If not, can you add these and report their impact on rare-event metrics?
> > > >
> > > >
> > > > Thank you for the insightful comment. In our main results (Tables 1 and 2), we did not apply explicit class-rebalancing. To address the concern regarding rare-event performance, we now include additional results in **Tables 8 and 9 in App Section A.11**, where we report F1-scores after applying class reweighting in the downstream classifier. These results provide a clearer view of model behavior on minority classes and complement the original metrics.

---

### Official Review · Reviewer_ciJy · 2025-10-31

**Soundness:** 2
**Presentation:** 3
**Contribution:** 2
**Rating:** 4
**Confidence:** 3

**Summary:**

The paper proposes Masked Multi-Scale Reconstruction, a method for PPG representation learning. In this approach, PPG signals are decomposed into wavelet coefficients using a discrete wavelet transform, which are then masked and subsequently reconstructed to learn meaningful representations. The model is trained on a large proprietary dataset and evaluated against several baselines across 13 downstream tasks.

**Strengths:**

- The idea of learning PPG representations through the reconstruction of masked DWT coefficients is quite interesting.
- The paper includes thorough experiments, with useful case studies and ablation analyses beyond standard downstream evaluations.
- The paper is well-written and easy to follow.

**Weaknesses:**

**Evaluation:** The diversity and number of devices used are essential for interpreting the results, and reporting these details would not compromise anonymity. However, the use of a closed-source dataset limits the interpretability and reproducibility of the findings. Specifically: (1) the number of datasets from which each downstream task is derived remains unclear, and (2) it is uncertain whether the training and test data originate from the same devices. Furthermore, several public PPG datasets [1] could be utilized for external validation to strengthen the evaluation.

This point is particularly important because, when the dataset is fixed and only the training method is varied (Table 2), the average performance improvement of the proposed approach is relatively small (\~1%) compared to open-source models trained on different data (\~3.5%). This suggests that, if the baselines were trained using the same data, their performance could be comparable to MMR, thereby weakening the claim of improved generalizability.

**Capture multi-resolution characteristics** such as: high-frequency transients, subtle waveform changes, and low-frequency rhythms such as respiration and circadian trend. Is there any supporting evidence or case study demonstrating that some of these characteristics are captured by MMR. Furthermore, many of the downstream tasks involve blood lab measurements, for which such multi-resolution features may not be directly relevant. This suggests a potential mismatch between the motivation of MMR and the choice of downstream tasks used for evaluation. In short, there could be a potential mismatch between the motivation of MMR and and the choice of downstream tasks used for evaluation.

**Experiments:** Many downstream tasks are strongly correlated with demographic variables. Therefore, it would be valuable to evaluate how the proposed approach performs on demographic prediction tasks such as age, sex, and BMI. Moreover, as the model is closed-source, these results and other relevant factors could be compared to the findings reported by Abbaspourazad _et al._ [2] for additional context.

Additionally, it would be insightful to examine the advantages of wavelet-based reconstruction through MMR compared to directly reconstructing the raw PPG signal. Such an experiment could help clarify the specific benefits of the wavelet decomposition stage and further strengthen the empirical analysis.

**Contribution beyond previous work:** While the proposed methodology is interesting, the contribution beyond prior work appears limited. (1) The approach is closed-source and comparable to [1], which leverages larger datasets to demonstrate that PPG representations can generalize to more than 50 downstream tasks. In [2], an open-source model is introduced using a morphology-aware self-supervised learning (SSL) strategy that generalizes across diverse tasks. Apart from the training framework, it is difficult to clearly distinguish this work in terms of experimental design, task selection, and case studies from previous research to justify publication. Moreover, the performance improvements of MMR are marginal compared to other SSL approaches trained on the same data. Considering these points, I lean toward a reject.

[1] Pillai, A., Spathis, D., Kawsar, F., & Malekzadeh, M. (2024). Papagei: Open foundation models for optical physiological signals. _arXiv preprint arXiv:2410.20542_.

[2] Abbaspourazad, S., Elachqar, O., Miller, A. C., Emrani, S., Nallasamy, U., & Shapiro, I. (2023). Large-scale training of foundation models for wearable biosignals. _arXiv preprint arXiv:2312.05409_.

**Questions:**

- What is the rationale behind using backbones of different sizes in Table 1? Can't MMR be matched to 5M parameters.
- Were other transforms beyond DWT considered for reconstruction?

---

> ### Author Response · Authors · 2025-11-24
> **Rebuttal**
>
> ### Weakness 1: Evaluation Diversity
> > Evaluation: The diversity and number of devices used are essential for interpreting the results, and reporting these details would not compromise anonymity. However, the use of a : closed-source dataset limits the interpretability and reproducibility of the findings. Specifically(1) the number of datasets from which each downstream task is derived remains unclear, and (2) it is uncertain whether the training and test data originate from the same devices. Furthermore, several public PPG datasets [1] could be utilized for external validation to strengthen the evaluation.
>
> > This point is particularly important because, when the dataset is fixed and only the training method is varied (Table 2), the average performance improvement of the proposed approach is relatively small (1%) compared to open-source models trained on different data (3.5%). This suggests that, if the baselines were trained using the same data, their performance could be comparable to MMR, thereby weakening the claim of improved generalizability.
>
> **Details on Pretraining/Downstream Dataset.** For pretraining, PPG data is collected from atleast 7 smartwatch versions (for some users the device metadata is unknown) and ~32k users. We have now added the device version versus user count  plot in  the revised manuscript in Fig. 8 in Appendix A.7. The distribution of devices across users exhibits a high variability. Our 19 downstream tasks are derived from 4 different cohort studies and one public dataset (DREAMT [1]). In Appendix A.7.1, we have added the number of users and number of segments in each dataset. All downstream experiments use user-level 5-fold cross-validation, so there is no subject overlap between train, validation, and test sets. We note that while users are disjoint across splits, device versions may still be shared between train and test.
>
> **Public Dataset Baseline.** To further validate our findings, we include 4 new tasks on sleep stage classification on DREAMT dataset. This dataset was collected from a cohort of 100 participants with and without sleep disorder with PPG smartwatch signals and expert sleep technician–annotated sleep stage labels. The results were added in Table 1 and 2 in revised manuscript where MMR improved over the Open source baseline (PaPaGei) and also on many other SSL baselines.
>
> **New Baseline- PaPaGei-Ours.** We agree that comparing to models trained on different data can conflate modeling and data-scale effects. We have updated the manuscript to include a baseline that retrains a state-of-the-art open-source baseline on our dataset (PaPaGei-Ours). This baseline is pretrained from scratch on our corpus. We follow the training procedure and model architecture of PaPaGei-S  as detailed in [1].  Please see Table 2 for detailed results. Given the strength of these baselines and the fact that they are trained on our data, we view the consistent improvements delivered by MMR as  the proposed multiscale DWT-based objective yields meaningfully better, more transferable representations.
>
> [1] Wang, Ke, et al. "DREAMT: Dataset for Real-time sleep stage EstimAtion using Multisensor wearable Technology." PhysioNet https://doi. org/10.13026/62AN-CB28 (2024).
>
> ### Weakness2:  Motivation/Task Mismatch
> > Is there any evidence that high-frequency transients / subtle waveform changes / low-frequency rhythms are actually captured? Many tasks are blood labs, where such features may not be directly relevant, suggesting a mismatch between motivation and tasks.
>
> We thank the reviewer for pointing out this detail. Our intention was to evaluate MMR on a broad set of tasks. Hypertension, BP and  PVC are the targets where beat morphology and short-term rhythms [1,2] are expected to benefited, and these are where we see the clearest relative gains. The blood lab tasks, by contrast, are included primarily as additional stress tests of transferability: they probe whether representations learned from short PPG segments can still provide useful signal for lab outcomes. We therefore view the lab results as complementary evidence of generalization, rather than the main justification for the wavelet design. We have also added age prediction tasks and 4 sleep staging tasks to expand our evaluation suite
>
> [1] Merino-Monge, Manuel, et al. "Heartbeat detector from ECG and PPG signals based on wavelet transform and upper envelopes." Physical and Engineering Sciences in Medicine 46.2 (2023): 597-608.
>
> [2] Yang, Chengming, et al. "Using PPG signals and wearable devices for atrial fibrillation screening." IEEE Transactions on Industrial Electronics 66.11 (2019): 8832-8842.

---

> ### Author Response · Authors · 2025-11-24
> **Rebuttal**
>
> ### Weakness 3: Additional Experiments
> > Experiments: Many downstream tasks are strongly correlated with demographic variables. Therefore, it would be valuable to evaluate how the proposed approach performs on demographic prediction tasks such as age, sex, and BMI. Moreover, as the model is closed-source, these results and other relevant factors could be compared to the findings reported by Abbaspourazad et al. [2] for additional context.”Experiments: Many downstream tasks are strongly correlated with demographic variables. Therefore, it would be valuable to evaluate how the proposed approach performs on demographic prediction tasks such as age, sex, and BMI. Moreover, as the model is closed-source, these results and other relevant factors could be compared to the findings reported by Abbaspourazad et al. [2] for additional context.”
>
>
>
> We thank the reviewer for this important task inclusion. In the revised manuscript, we add **two demographic downstream tasks** on our cohort: age regression and age-group classification (Sec. 5, Tabs. 1–2). Using the same pipeline as for other tasks, MMR achieves the lowest MAE for age regression and the highest AUROC of 78.2 for age classification, outperforming self-supervised baselines including SimCLR, MSN, TF-C, PaPaGei, and open source baselines. Within our controlled setting where all methods are trained and evaluated on the same free-living wearable cohort our multiscale MMR representations provide the best demographic performance.
>
>
> > Additionally, it would be insightful to examine the advantages of wavelet-based reconstruction through MMR compared to directly reconstructing the raw PPG signal. Such an experiment could help clarify the specific benefits of the wavelet decomposition stage and further strengthen the empirical analysis.
>
> We agree that the benefit of the wavelet decomposition should be isolated. In the revised version, we add a time-domain masked autoencoder baseline, **Masked Time Reconstruction (MTR)**, which uses the same ViT encoder–decoder, masking ratio, patch size, optimizer, training schedule, and pretraining data as MMR, but operates on patchified raw PPG signals and reconstructs the waveform instead of DWT coefficients (Sec. 5.2, Tab. 1). MMR (DWT) improves performance over MTR on 17/19 tasks, with gains on waveform- and rhythm-sensitive endpoints such as free-living hypertension, PVC detection. With architecture and data held fixed, these results show that the wavelet-based multiscale reconstruction leads to performance gains over what is achieved through reconstruction using  raw PPG signal

---

> ### Author Response · Authors · 2025-11-25
> **Rebuttal**
>
> ### Weakness 4: Contribution Beyond Existing Works
>
> > Contribution beyond previous work: While the proposed methodology is interesting, the contribution beyond prior work appears limited. (1) The approach is closed-source and comparable to [1], which leverages larger datasets to demonstrate that PPG representations can generalize to more than 50 downstream tasks. In [2], an open-source model is introduced using a morphology-aware self-supervised learning (SSL) strategy that generalizes across diverse tasks. Apart from the training framework, it is difficult to clearly distinguish this work in terms of experimental design, task selection, and case studies from previous research to justify publication. Moreover, the performance improvements of MMR are marginal compared to other SSL approaches trained on the same data. Considering these points, I lean toward a reject.
> [1] Pillai, A., Spathis, D., Kawsar, F., & Malekzadeh, M. (2024). Papagei: Open foundation models for optical physiological signals. arXiv preprint arXiv:2410.20542.
> [2] Abbaspourazad, S., Elachqar, O., Miller, A. C., Emrani, S., Nallasamy, U., & Shapiro, I. (2023). Large-scale training of foundation models for wearable biosignals. arXiv preprint arXiv:2312.05409.
>
> We thank the reviewer for this detailed perspective and for highlighting [1,2]. We discuss the positioning of MMR with respect to these works in the following.
>
> **Novel DWT Formulation.** We provide a detailed discussion on why a wavelet-based transform is well-suited for modelling physiological biosignals such as PPG in lines 815-829 of App A.3. We hope this discussion clarifies our contribution.
>
> **Comparison to Abbaspourazad et al [2]**. This work does not present any results of their method on public PPG evaluation datasets. In MMR we present evaluation across a suite of 19 downstream tasks spread across 4 datasets collected from atleast 7 distinct device versions (See App A.7 for details). Importantly, we provide results on sleep stage classification on a public domain dataset (DREAMT [3]) and find that MMR generally outperforms competitive baselines (see Table 1,2 of updated manuscript). Additionally, we provide comparison to multiple state-of-the-art foundation models (such as PaPaGei and Chronos).
>
> **Comparison to PaPaGei [1].** In Table 2 of manuscript, we provide a neck-to-neck comparison with PaPaGei and find that MMR outperforms the zero-shot PaPaGei model on 16 out of 19 evaluated downstream tasks. Additionally, for a more fair comparison we also train the PaPaGei model from scratch on our proprietary dataset (PaPaGei-Ours) and observe that MMR still outperforms this baseline. We attribute these gains in performance of MMR to our novel wavelet-based PPG signal representation which enables capturing fine-grained, multi-resolution characteristics (see lines 815-829 in App A.3 for detailed discussion). Furthermore, we leverage a masked reconstruction objective which performs better than the contrastive learning on our downstream tasks. This is also evidenced by the performance gains of the time-domain masked reconstruction baseline (MTR) that we train on our dataset (See Table 1) over PaPaGei-Ours.
>
>
> [1] Pillai, A., Spathis, D., Kawsar, F., & Malekzadeh, M. (2024). Papagei: Open foundation models for optical physiological signals. arXiv preprint arXiv:2410.20542.
>
> [2] Abbaspourazad, S., Elachqar, O., Miller, A. C., Emrani, S., Nallasamy, U., & Shapiro, I. (2023). Large-scale training of foundation models for wearable biosignals. arXiv preprint arXiv:2312.05409.”
>
> [3] Wang, Ke, et al. "DREAMT: Dataset for Real-time sleep stage EstimAtion using Multisensor wearable Technology." PhysioNet https://doi.org/10.13026/62AN-CB28 (2024).
>
> ### Questions:
>
> **Backbone sizes / 5M parameters**
>
> Many baselines have varied parameter count (SimCLR is 5M, TF-C is 10 M ). We picked a ViT encoder parameter as used by [1] as one of the small variants for their reconstruction task.  We have included both a 2M encoder which is competitive on many tasks and is best/second best on many tasks. We expect a 5M encoder to perform at a similar range.
>
> [1] Narayanswamy, Girish, et al. "Scaling wearable foundation models." arXiv preprint arXiv:2410.13638 (2024).
>
> **Other transforms beyond DWT**
>
> In **App A.3  we provide a detailed discussion on the benefits of employing a wavelet-based transformation** since it provides a multi-scale, spatially localized representation. We show consistent improvements on multiple competitive baselines on a diverse task suite using this approach (see Table 1,2). In the current work, we did not systematically evaluate alternative transforms and we view the exploration of other analytic transforms as an interesting and complementary direction for future work.

---

### Official Review · Reviewer_z2s9 · 2025-11-05

**Soundness:** 3
**Presentation:** 3
**Contribution:** 3
**Rating:** 6
**Confidence:** 4

**Summary:**

This paper introduces MMR, a self-supervised pretraining framework for large-scale PPG foundation models. The approach decomposes PPG signals into hierarchical time–frequency representations via discrete wavelet transforms and trains a ViT-based encoder–decoder to reconstruct masked coefficients across scales. The method uses PPG segments from smartwatch users collected in the field. The pretrained representations are evaluated on 13 downstream cardiovascular and biomarker prediction tasks, where MMR strong results comparing with the baselines. Ablations explore the impact of these different components on the performance of the model.

I really like the future direction of exploring adaptive multiscale strategies.

**Strengths:**

1. The use of wavelet-based multiscale reconstruction as a masked modeling target is a strong conceptual contribution.
2. The diversity of the dataset, with 1h30 of data for each patient in unconstrained environments significantly increases applicability over prior foundation models trained on clean, clinical datasets.
3. The data pre-processing has a good balance between cleaning and maintaining as much data as possible.
4. The paper benchmarks across 13 diverse downstream tasks (clinical and physiological), both classification and regression.
5. Well presented ablation study. Embedding visualizations (t-SNE, silhouette) convincingly show physiological structure capture.
6. MMR-Light variant maintains strong performance with few parameters.
7. Paper is well written and easy to read. (except figures/tables, see below)

**Weaknesses:**

1. While effective, MMR’s novelty lies mainly in applying masked reconstruction to wavelet coefficients. The method reuses a ViT backbone with minimal architectural innovations.
2. Key preprocessing and DWT hyperparameters (e.g., sampling-rate normalization, interpolation scheme for coefficients) could be better detailed for reproducibility.
3. The baselines could be newer models (eg Chronos-Bolt instead of Chronos)
4. Clarity on the fixed parameters in the ablation study could be improved (what model size was used for the data scaling ablation and conversely what data size for the model scaling?
5. It is disputable if this is indeed a diverse population with "high variability" as the smart-watch carrying population is rather restricted.
6. The min-max range is very large, with 10-15% of the value and the models only being within a couple percentage points of each-other.

Clarity/formatting:
- Figure 2 axis titles are hard to read
- The tables could be improved for readability (grouping T.1+T.2, making a graph, etc.)

**Questions:**

1. Could MMR generalize to other biosignals (e.g., ECG, accelerometer)? Does the wavelet-based representation transfer across modalities?
2. How do performance trends change if downstream tasks are fine-tuned rather than frozen?
3. Were any measures taken to mitigate potential data leakage across users or sessions?
4. How does computational cost compare in terms of FLOPs and convergence speed?
5. Was any consideration given to using a diffusion model?
6. Have the baseline models (eg. Chronos) been finetuned on this task/with this data? Do you think this would be relevant considering the small performance gap between it and MMR?
7. How does the model's ability to isolate individual patients with higher patient-wise distance affect it's out-of-distribution abilities? Especially on new patients?
8. Any intuition on why different models performs as they do on the different tasks, as per model task-performance rankings are not consistent?

---

> ### Author Response · Authors · 2025-11-24
> **Rebuttal**
>
> ### Weakness 1 Minimal Architectural Innovation
> > While effective, MMR’s novelty lies mainly in applying masked reconstruction to wavelet coefficients. The method reuses a ViT backbone with minimal architectural innovations.
>
> We thank the reviewer for their feedback. Our backbone is intentionally standard. We do not claim architectural novelty at the ViT level; our contribution is a different inductive bias and pretraining objective for large scale PPG training: a wavelet-based, multiscale masked reconstruction framework. MMR introduces (i) a DWT band×time tokenizer aligned with physiological frequency ranges and (ii) a cross-scale masking/reconstruction objective that explicitly couples low- and high-frequency bands. To show that this is not a cosmetic change, we add a time-domain MAE baseline (MTR) (in Table 1)  with the same ViT encoder–decoder, masking ratio, optimizer, schedule, and data; MMR outperforms MTR on 17/19 tasks. By holding the architecture and data fixed, the experiments directly attribute the gains to the proposed wavelet-based inductive bias rather than to new network components. Furthermore, we outperform a state-of-the-art, open-source baseline which uses the exact same pretrained data and downstream evaluation protocol as MMR (see PaPaGei-Ours in Table 2)
>
>
> ------------
>
> ### Weakness 2: Missing Details
> > Key preprocessing and DWT hyperparameters (e.g., sampling-rate normalization, interpolation scheme for coefficients) could be better detailed for reproducibility.
>
> In the revision, we add a clear description of the preprocessing and DWT hyperparameters in App. A.6.1. We resample all segments rates to 100 Hz and apply zero-order interpolation to get the 2D cofficient map. These additions fully specify the preprocessing and DWT choices needed for reproducibility.
>
>
> ------------
>
> ### Weakness 3:  Newer Baselines
> > The baselines could be newer models (eg Chronos-Bolt instead of Chronos)”
>
> We have added results with **Chronos-Bolt as an additional baseline** in Appendix A.5.3 and  find that MMR consistently performs better than both Chronos-Bolt/Chronos. However, the newer variant has higher performance compared to Chronnos but still lag behind the PPG specific baselines. Please see **App A.5.3** for detailed discussion results.
>
>
> ------------
>
> ### Weakness 4:  Clarification on Fixed Parameters
> > Clarity on the fixed parameters in the ablation study could be improved (what model size was used for the data scaling ablation and conversely what data size for the model scaling?”
>
> In our model scaling ablation, we fix the pretraining data size to 17M segments and vary the model size (1M, 2M, 7M parameters). In our data scaling ablation, we fix the model size to 7M parameters and vary the amount of pretraining data (1M, 5M, 17M segments). We now state these fixed choices explicitly in Sec. 5.3  to make the ablation setup clearer.
>
>
> ------------
>
> ### Weakness 5: Scope of Study
> > It is disputable if this is indeed a diverse population with "high variability" as the smart-watch carrying population is rather restricted.
>
> We agree that smartwatch users are not fully representative of the global population. In the revised manuscript, we tone down our phrasing (lines 23-24) to remove claims around population-level representativeness. Our goal is to study foundation models in the realistic setting of consumer wearables, and we now explicitly state this scope and limitation in the text.
>
>
> ------------
>
> ### Weakness 6: Performance Min-Max Range
>
> > The min-max range is very large, with 10-15% of the value and the models only being within a couple percentage points of each-other.
>
> We agree that some of the min–max ranges appear large relative to the mean. This is a consequence of our user-level 5-fold cross-validation on noisy, generally free-living data: different folds contain different subsets of users, so absolute AUROC can vary. This is common across all methods. App. A.6.4 further expands on this evaluation setup. A similar pattern of wide confidence intervals across folds is also reported in prior PPG foundation-model work [1], reflecting the difficulty and heterogeneity of these clinical endpoints across users.
>
> [1] PaPaGei: Open Foundation Models for Optical Physiological Signals, ICLR 2025

---

> ### Author Response · Authors · 2025-11-24
> **Rebuttal**
>
> ### Question1 Generalization of MMR to other Biosignals
> > Q1 Could MMR generalize to other biosignals (e.g., ECG, accelerometer)? Does the wavelet-based representation transfer across modalities?”
>
> Architecturally, MMR is modality-agnostic at the encoder level: it only assumes a 2D band×time map and does not rely on PPG-specific features. In principle, any 1D biosignal (e.g., ECG, accelerometer) that admits a meaningful wavelet decomposition could be plugged into the same pipeline by adapting only the front end (sampling rate, passband, wavelet family/levels). Wavelet representations are already used for other signal **ECG (e.g., QRS detection and HRV/arrhythmia analysis) [1]** which supports the plausibility of this multiscale inductive bias across modalities. However, in this work we focus on large-scale PPG, and we do not claim empirical cross-modal results. Extending MMR to other biosignals and systematically evaluating transfer is an interesting direction for future work.
>
> [1] Addison, Paul S. "Wavelet transforms and the ECG: a review." Physiological measurement 26.5 (2005): R155.
>
>
> ------------
>
> ### Question2 Finetuning Trends
>
>
> > Q2 How do performance trends change if downstream tasks are fine-tuned rather than frozen?
>
> In this work we follow common practice in the foundation-model literature [1,2,3] and primarily evaluate frozen encoders with linear probes, which directly measures representation quality and keeps comparisons across methods clean and reproducible. **Since MMR already shows consistent gains over a broad set of strong baselines under this controlled probe setting, we expect the relative trends to largely persist under task-specific fine-tuning.**
>
> [1] PaPaGei: Open Foundation Models for Optical Physiological Signals, ICLR 2025
>
> [2] Large-Scale Training Of Foundation Models For Wearable Biosignals, ICLR 2024
>
> [3] Pulse-PPG: An Open-Source Field-Trained PPG Foundation Model for Wearable Applications Across Lab and Field Settings, ACM IMWUT 2025
>
>
> ------------
> ### Question3 Mitigation of  potential data leakage
>
> > Q3 Were any measures taken to mitigate potential data leakage across users or sessions?”
>
> Yes, our downstream setup is explicitly designed to avoid leakage. For all supervised downstream tasks, we use **user-level 5-fold cross-validation.** All segments from a given user (and all their sessions) are assigned to a single fold. There is no user overlap between users across train, validation, and test splits. We also ensure that data for users in pretraining splits are not used for downstream evaluation. More details added in **Appendix Section A.6.4.**
>
>
> ------------
> ### Question4  Computational Cost
> > Q4. How does computational cost compare in terms of FLOPs and convergence speed?”
>
> MMR and the time-domain baseline (MTR), use the same ViT backbone, batch size, optimizer, and training schedule, so the FLOPs per step and number of steps to convergence are comparable by design. The only additional cost in MMR is the DWT-based front-end, which is a fixed, lightweight preprocessing step; in practice, this adds only a small constant overhead relative to the transformer encoder computation. Empirically, Papagei, MMR-Light, MTR, and MMR all operate in the sub-1 GFLOPs regime (0.120, 0.148, 0.217, and 0.424 GFLOPs, respectively), while Chronos is substantially more expensive in our setup (96.84 GFLOPs).
>
>
> ------------
> ### Question5  Diffusion Model
> Q5. Was any consideration given to using a diffusion model?
>
> In this work we focused on masked reconstruction with a transformer backbone which is a dominant pretraining paradigm and is particularly convenient for isolating the effect of our proposed wavelet-based multiscale representation. We did not explore diffusion-based objectives, which would introduce substantially different training dynamics and design choices (e.g., noise schedules, denoising architectures) and make it harder to attribute gains specifically to the DWT-based inductive bias. We view time-series diffusion models as a complementary direction, and adapting MMR’s wavelet tokenizer to a diffusion framework is an interesting avenue for future work, but it is beyond the scope of the present paper.

---

> > ### Author Response · Authors · 2025-11-24
> > **Rebuttal**
> >
> > ### Question6 Finetuning Baseline Methods
> >
> > > Q6. Have the baseline models (eg. Chronos) been finetuned on this task/with this data? Do you think this would be relevant considering the small performance gap between it and MMR?”
> >
> > All self-supervised baselines (TF-C, SimCLR) and the new baseline (MTR - Masked Time Reconstruction) are trained on the same pretraining data as MMR. To more fairly control for pretraining data, we added another new baseline, PaPaGei-Ours (5M), where we retrain the state-of-the-art, open-source PaPaGei [1] architecture from scratch on our pretraining data using the same preprocessing, pretraining schedule, and evaluation pipeline as MMR. In this strictly same-data setting, MMR and MMR-Light consistently outperform PaPaGei-Ours (and the other SSL baselines) by 5-10/% on some tasks (see Table 1,2 in paper for detailed breakdown), indicating that MMR’s gains are not solely due to differences in pretraining data, and that the wavelet-based objective provides a genuine advantage beyond what is achieved by existing general time-series models on our PPG-based task suite. Chronos is used in our work as an off-the-shelf general time-series foundation model and is not fully finetuned. Training a large scale (200M) is a computationally expensive routine and we follow evaluation protocols from prior work [1,2,3] which conduct linear probing on frozen embeddings of large-scale foundation models.
> >
> >
> > ------------
> >
> >
> >
> > ### Question7 Out of Distribution Abilities
> > > Q7. How does the model's ability to isolate individual patients with higher patient-wise distance affect it's out-of-distribution abilities? Especially on new patients?”
> >
> > In Fig. 2(a) of Section 5.4,  we show the inter patient distance in the embeddings obtained from MMR vs other methods in Hypertension Free Living dataset. This figure depicts that instead of collapsing all patients embedding together our encoder learns broader features across patients and learns more discriminability. However, since all reported performance in our evaluation suite  is  computed under strict user-level splits that is held-out /unseen patients at test time) the performance of MMR shows that capturing patient-specific nuances helps, rather than hurts, generalization to new individuals, in other words, it improves out-of-distribution performance at the patient level.  A similar pattern of inter-patient embeddings being a good indicator for capturing broad features was also noted in [1]
> >
> >
> > ------------
> >
> > ### Question 8 Varied Model Task Performance
> > Q8. Any intuition on why different models performs as they do on the different tasks, as per model task-performance rankings are not consistent?”
> > .
> > We agree that rankings are not identical across all tasks, but we do observe a clear overall pattern. Our task suite is deliberately heterogeneous: waveform- and rhythm-driven endpoints (free-living hypertension, BP, PVC, sleep staging) rely strongly on beat morphology and short-term dynamics, and on these tasks MMR/MMR-Light are consistently the top performers. For several lab targets, all methods show variable relative rankings, with different models winning on different tasks. However, over the full 19-task suite, MMR achieves strong performance with masked time recognition that the wavelet-based multiscale representation is systematically stronger even if rankings on a few individual tasks are not perfectly monotone. However, the heterogeneity of downstream tasks and user cohorts can lead to varied model-task performance trends.
> >
> > [1] PaPaGei: Open Foundation Models for Optical Physiological Signals, ICLR 2025
> > [2] Large-Scale Training Of Foundation Models For Wearable Biosignals, ICLR 2024
> > [3] Pulse-PPG: An Open-Source Field-Trained PPG Foundation Model for Wearable Applications Across Lab and Field Settings, ACM IMWUT 2025

---

### Official Review · Reviewer_kLnH · 2025-11-05

**Soundness:** 3
**Presentation:** 3
**Contribution:** 2
**Rating:** 4
**Confidence:** 5

**Summary:**

This paper introduces a new self-supervised pretraining framework for PPG foundation models, which applies Discrete Wavelet Transform (DWT) to decompose raw PPG into multiple frequency bands, then trains a Vision Transformer encoder-decoder to reconstruct randomly masked wavelet coefficients. The authors pretrain on approximately 17 million 10-second PPG segments from over 32,000 smartwatch users and evaluate on 13 diverse health prediction tasks, including hypertension detection, arrhythmia classification, and blood biomarker prediction. The results show that MMR achieves competitive performance with existing PPG foundation models.

**Strengths:**

This paper tests 13 different downstream tasks spanning cardiovascular conditions (hypertension, PVC), metabolic markers (creatinine), and electrolyte imbalances, and provides strong evidence that the learned representations capture broadly useful information.

**Weaknesses:**

1. The core technical contribution lacks clear validation. While wavelets are positioned as the main innovation, the paper never isolates whether DWT actually drives the performance gains. Critically, the paper is missing the essential ablation: MMR with DWT versus MMR without DWT—the same masked autoencoder architecture and training procedure applied to patchified raw PPG time series instead of wavelet coefficients.

2. Results are mixed and claims are overstated. The abstract and conclusions describe MMR as achieving "superior" performance that "outperforms" baselines, but the actual results tell a more nuanced story.

3. A fundamental limitation undermines the foundation model premise. The ablation studies reveal a critical tension: different tasks benefit from different wavelet decomposition depths. Hypertension classification works best with level 3, while lab biomarkers like WBC and Sodium prefer level 5. This is not a minor implementation detail—it directly contradicts the paper's core premise of learning "robust, transferable features" that generalize across diverse tasks. The authors acknowledge this as a "key limitation" and suggest exploring "adaptive multiscale strategies" in future work, but this limitation is severe enough to undermine the approach's viability.

4. The "multi-scale" framing oversells temporal coverage. The paper claims to capture features "spanning fine-grained waveform morphology to global rhythmic dynamics" and explicitly mentions "circadian trends" (line 81). However, the 10-second window fundamentally cannot capture true long-term physiological scales. The lowest frequency reliably analyzable in 10 seconds is approximately 0.1 Hz—anything slower, including the circadian rhythms mentioned, is completely inaccessible. What the model actually learns are multi-resolution representations within short windows: beat morphology (~1-10 Hz) to respiratory and brief heart rate variability patterns (~0.1-1 Hz).

**Questions:**

1. Could you explain how variable sampling rates (25-100 Hz) are handled, despite this being essential for interpreting the wavelet decomposition.

2. Do the same coefficient positions correspond to the same physiological phenomena across different sampling rates, or does rate variation change the meaning of features?

3. Will you release the weights of the pre-trained model and datasets used from pre-training and downstream tasks for reproducibility? The current datasets come from proprietary smartwatch data that cannot be disclosed, rendering exact reproduction impossible.

**Details Of Ethics Concerns:**

This paper mainly uses unpublished datasets and does not disclose the data collection process.

---

> ### Author Response · Authors · 2025-11-24
> **Rebuttal**
>
> ### Weakness 1: Missing Time Reconstruction Baseline
> > *The core technical contribution lacks clear validation. While wavelets are positioned as the main innovation, the paper never isolates whether DWT actually drives the performance gains. Critically, the paper is missing the essential ablation: MMR with DWT versus MMR without DWT—the same masked autoencoder architecture and training procedure applied to patchified raw PPG time series instead of wavelet coefficients.*
>
> We thank the reviewer for pointing out this important ablation and fully agree that isolating the effect of the wavelet front-end is essential. In the revised manuscript, we add this comparison: Masked Time Reconstruction (MTR), a time-domain masked autoencoder that uses the same ViT encoder–decoder, masking ratio, patch size, optimizer, training schedule, and pretraining data as MMR, but operates on patchified raw PPG and reconstructs the waveform instead of DWT coefficients **(Sec. 5, Tab. 1 and Appendix A.5.2).**
>
> This directly addresses “**MMR with DWT vs. MMR without DWT.**” Across 19 downstream tasks, MMR (DWT) improves performance over MTR on 17/19 tasks, notably on key cardiovascular endpoints (free-living hypertension, PVC), demographics (age regression and classification), sleep staging on a new public dataset [1] (see Table 1,2 of paper). With architecture, data, and training held fixed, changing only the representation and reconstruction target (time vs. multiscale wavelet coefficients) improves performance, demonstrating that the DWT front-end is a driver of gains for large-scale PPG pretraining. We have also updated the manuscript with discussion on this crucial ablation in Sec. 5.1.
>
> ------------
>
>
> ### Weakness 2: Results are overstated:
> > Results are mixed and claims are overstated. The abstract and conclusions describe MMR as achieving "superior" performance that "outperforms" baselines, but the actual results tell a more nuanced story.
>
> We thank the reviewer for this observation and have toned down the claims in the revised manuscript (see revised Sec 5.1). We also provide detailed clarification on additional task evaluation in the following and acknowledge task settings where MMR lags behind baselines.
>
> **MMR performance across tasks.**  Tables 1–2 evaluate a deliberately broad and challenging suite of 19 classification tasks spanning cardiovascular health and sleep staging. We also add 2 tasks focused on demographics (age analysis) on our evaluation dataset. In the table below, we also provide additional evaluation results on 4 tasks sourced from publicly available evaluation dataset (DREAMT). These additional tasks include sleep staging classification. Across this heterogeneous set, MMR and its lightweight variant MMR-Light show strong performance: MMR attains the highest average AUROC (in Table 1 and 2) and is best or tied on 17/19 tasks with many self-supervised baselines.
>
> **Where MMR Lags.**  For the few lab endpoints where MMR does not clearly win (e.g., Creatinine, Hemoglobin), performance across methods is not consistent, underscoring the challenging nature of mapping PPG signals to lab-report outcomes.
>
> ------------

---

> ### Author Response · Authors · 2025-11-24
> **Rebuttal**
>
> ### Weakness 3 : Decomposition Dependent  Task Performance
> > A fundamental limitation undermines the foundation model premise. The ablation studies reveal a critical tension: different tasks benefit from different wavelet decomposition depths. Hypertension classification works best with level 3, while lab biomarkers like WBC and Sodium prefer level 5. This is not a minor implementation detail—it directly contradicts the paper's core premise of learning "robust, transferable features" that generalize across diverse tasks. The authors acknowledge this as a "key limitation" and suggest exploring "adaptive multiscale strategies" in future work, but this limitation is severe enough to undermine the approach's viability.
>
> We thank the reviewer for raising this point. We agree that understanding how different tasks interact with multilevel decomposition structure is important. However, in our main experiments, a single MMR model is pretrained once, with a fixed wavelet family and decomposition level, on 7M PPG segments and then reused across all 19 downstream tasks. **As shown in the revised Tables 1 and 2, this fixed configuration consistently achieves strong performance, across varied tasks. These transferable features are learnt without any per-task tuning of decomposition depth.**
>
>
>
> **MMR decomposition depth ablation.** We would like to note that the depth ablation is conducted with a smaller model (2 Mil parameters) pretrained on a 1M-sample subset (Section. 5.4.).  where more exhaustive sweeps are feasible; we have clarified this distinction in the revision (Section. 5.4.). These results indicate that decomposition depth can affect downstream performance, but we found that training on a large-scale corpus with moderate depths already provides strong performance across endpoints, and a single fixed-depth MMR remains a viable and effective foundation model for PPG representation learning. During the rebuttal we re-trained the full MMR model (7 Mil) on the entire pretraining dataset using three levels: L3, L4 and L5. From the results, we found L3 to work best across and have updated numbers in revised manuscript. However, we view more principled strategies for adaptively choosing the decomposition level as an exciting direction for future work that could further enhance wavelet-based large-scale training for PPG foundation modeling.
>
> ------------
>
>
> ### Weakness 4: Temporal coverage
> > The "multi-scale" framing oversells temporal coverage. The paper claims to capture features "spanning fine-grained waveform morphology to global rhythmic dynamics" and explicitly mentions "circadian trends" (line 81). However, the 10-second window fundamentally cannot capture true long-term physiological scales. The lowest frequency reliably analyzable in 10 seconds is approximately 0.1 Hz—anything slower, including the circadian rhythms mentioned, is completely inaccessible. What the model actually learns are multi-resolution representations within short windows: beat morphology (1-10 Hz) to respiratory and brief heart rate variability patterns (0.1-1 Hz).
>
> We thank the reviewer for this careful observation and agree that our wording around "global rhythmic dynamics” and “circadian trends” was too broad given the 10 s pretraining window. Our model operates on 10s PPG segments  so the multiscale structure we capture is inherently short-term: from beat-level morphology  to lower-frequency components within that band that reflect respiration and short-term heart rate variability.  In the revision, we have tightened the framing accordingly. Specifically, we remove the explicit reference to “circadian trends” and have updated the manuscript with these changes (lines 78-80).
>
> ------------

---

> > ### Author Response · Authors · 2025-11-24
> > **Rebuttal**
> >
> > ### Questions:
> >
> > > Question1: Could you explain how variable sampling rates (25-100 Hz) are handled, despite this being essential for interpreting the wavelet decomposition ?Question1: Could you explain how variable sampling rates (25-100 Hz) are handled, despite this being essential for interpreting the wavelet decomposition ?
> >
> > PPG is collected from multiple smartwatch models with sampling rates between 25 and 100 Hz. Continuous streams are split into non-overlapping 10s segments, and each segment is resampled to a common 100 Hz grid using polyphase resampling, so every segment has length (T = 1000). We apply DWT on these resampled signals of fixed length across devices. We have added full implementation details of 2D coefficient map creation in Appendix A.6.1.
> >
> > > Question 2: Do the same coefficient positions correspond to the same physiological phenomena across different sampling rates, or does rate variation change the meaning of features?”
> >
> > As all segments are resampled to 100 Hz, the DWT operates under the same nominal sampling rate. As a result, the wavelet decomposition is defined with respect to a fixed grid: each row of the coefficient map corresponds to the same approximate frequency band, and each column to the same relative time offset within the 10 s window, regardless of the original device sampling rate. However, we retain only those wavelet subband rows whose frequencies lie within 0.5–10 Hz bandpass range, discarding higher-frequency subbands that can not be reliably represented from lower sampling rate devices. We provide details on data preprocessing in Appendix A.6.1 of the updated manuscript.
> >
> >
> > > Question3: Will you release the weights of the pre-trained model and datasets used from pre-training and downstream tasks for reproducibility? The current datasets come from proprietary smartwatch data that cannot be disclosed, rendering exact reproduction impossible.
> >
> > We recognise the value of open source code, releasing pretrained models for the research community and reporting reproducible evals. We discuss all of these in the following.
> >
> > **Open-sourcing Model/Data Details.** Due to privacy and institutional restrictions we are unable to release the pretrained weights or datasets used for pretraining. In an attempt to maximize reproducibility, we do provide all detailed hyperparameters for the MMR model architecture (Appendix A.6.2), detailed preprocessing pipeline (Appendix A.6.1) and specifics about pretraining datasets (Appendix A.7). We strongly believe that all aforementioned details and our comprehensive evaluations and ablations will provide crucial research and engineering insights to the community to build future foundation models for physiological biosignals. We are ready to provide further details to the best of our ability based on the reviewer’s response.
> >
> > **Public PPG Dataset Evals/New Tasks.** We have added public dataset evaluations with the pretrained MMR model to provide greater insights into the effectiveness of our method. For sleep staging, we include four tasks on the public DREAMT[1] dataset (Table 1 and Table 2).
> >
> > **Comparison to existing Open Source methods.** To strengthen comparability under identical data conditions, we also pretrain the open-source PaPaGei[2] method on our proprietary corpus (PaPaGei-Ours) using the same preprocessing and fine-tuning setup as MMR, and report these results in Table 2. This controls for differences in data scale and shows that MMR’s gains are not merely due to access to a larger or different dataset, but to the proposed multiscale wavelet-based objective.
> >
> > [1] Wang, Ke, et al. "DREAMT: Dataset for Real-time sleep stage EstimAtion using Multisensor wearable Technology." PhysioNet https://doi. org/10.13026/62AN-CB28 (2024).
> >
> > [2] PaPaGei: Open Foundation Models for Optical Physiological Signals, ICLR 2025

---

### Author Response · Authors · 2025-11-24
**General Response**

We thank all reviewers for their detailed and helpful feedback and are encouraged that they recognize the strengths of our work.

Reviewer **kLnH** notes that our method generates *"learned representations that capture broadly useful information across 13 tasks"*. Reviewer **z2s9** states that *"using wavelet masking as a modeling target is a strong conceptual contribution"*, and Reviewer **ciJy** also finds the *"reconstruction of masked DWT coefficients particularly interesting"*. Reviewer **H7f6** also highlights our treatment of PPG in the time-frequency domain as a strength due to natural alignment with *"physiological non-stationarity and multi-scale rhythms"*.

All reviewers unanimously appreciated our broad task evaluation and extensive ablation studies. In particular, Reviewer **ciJy** notes that the paper includes *"thorough experiments, with useful case studies and ablation analyses beyond standard downstream evaluations"*, and Reviewer **z2s9** praises the *"well-presented ablation study"* and the embedding visualizations (t-SNE, silhouette), which convincingly demonstrate that our *"representations capture meaningful physiological structure"*. Reviewer **H7f6** notes that *"ablations and scaling analyses offer practical guidance"*.

--------
## Summary of Changes

We are grateful for these positive comments. Based on the reviewers’ thoughtful suggestions and feedback, we have revised our manuscript significantly. All modified text is highlighted in blue, and all updated figures and tables have blue captions. We respond to individual reviewers’ questions and comments directly below their respective reviews. Here, we present the key changes to the manuscript:

- Addition of **6 New Downstream Evaluation Tasks** (Sec 4; Table 1,2):
	- 4 of them are sleep stage classification tasks to show evaluations on a public downstream dataset and include more multi-resolution target tasks
	- 2 age prediction tasks: binarised age classification and regression on our internal dataset to show evaluations on demographic prediction tasks
- Addition of **New Baselines** (Sec 5.1; Table 1,2):
	- **Masked Time Reconstruction (MTR):** This baseline uses same ViT encoder-decoder architecture, training paradigm and data as MMR but reconstructs patchified  raw PPG time series to isolate the effect of our proposed wavelet coefficient reconstruction. Further discussion App A.5.2
	- **PaPaGei-Ours:** We train the PaPaGei [1] model trained on our proprietary data to evaluate  results on downstream tasks in a comparable manner
	- **Chronos-Bolt:** Newer version of Chronos [2] has been added as an additional variant of a foundation time series model (Appendix Table 3)
- Updated **MMR (7 Mil)** result based on decomposition-depth ablation (Sec 3; Table 1,2)
- Additional **discussion on results** in Sec 5.1 and App Section. A.5
- **Comparison to frequency-aware MAE** methods in App Sec A.3
- Details for easier **reproducibility**:
	- Data preprocessing in App Section A.6.1
	- Pretraining dataset/evaluation protocol in App A.7 and A.6.4
	- Reproducibility/Ethics statements in App A.1 and A.2


[1] PaPaGei: Open Foundation Models for Optical Physiological Signals, ICLR 2025

[2] Chronos: Learning the Language of Time Series, TMLR 2024

---

> ### Author Response · Authors · 2025-11-28
> **Follow Up On Rebuttal Discussion**
>
> Dear reviewers,
>
> Thank you again for the detailed feedback and constructive suggestions. We have tried our best to carefully address the concerns raised and incorporate the requested updates and additional experiments, which we believe have strengthened the manuscript.
>
> As the author–reviewer discussion period is nearing its close, we would appreciate it if you could share whether key concerns remain after reviewing our rebuttal and the revised submission. If there are remaining questions, we are happy to use the remaining time to provide clarifications or run additional analyses as needed.
>
> The revised manuscript is available with tracked changes highlighted in blue.
>
> Thank you for your time and consideration.

---

### Author Response · Authors · 2025-12-03
**Summary of Reviews/Responses for New Area Chair**

In light of the discussion closure, we provide a summary of the reviewers’ comments and the changes we made during rebuttal to address all reviewer concerns. In the manuscript, the text in blue highlights all the changes.

------------


## Strengths

*Conceptually strong and interesting pretraining objective:* Wavelet-based multiscale masked reconstruction aligns naturally with the non-stationary, multi-rhythm nature of PPG signals and is noted as conceptually strong and interesting pretraining objective (Reviewers **z2s9, ciJy** and **H7f6**)

*Broad downstream effectiveness:* Performance benchmarking across diverse and broad  physiological tasks (Reviewer **kLnH** and **z2s9**)

*Thorough experimentation:* Comprehensive ablations beyond standard evaluations, scaling studies, and embedding visualizations (Reviewers **ciJy**, **z2s9** and **H7f6**)

*Model efficiency options:* The MMR-Light variant retains competitive performance with fewer parameters (Reviewer **z2s9**)

------------


## Addressed Weaknesses

### **New Baselines**
*Masked Time Reconstruction (MTR)* (Reviewer **kLnH**, **ciJy**). We added MTR that uses the same architecture, training setup and pretraining data as Masked Multiscale Reconstruction (MMR), but reconstructs patchified raw PPG waveform. We find that MMR improves over MTR which isolates the impact of wavelet-based reconstruction (Table 1; App A.5.1).

*PaPaGei-from-scratch* (Reviewer **z2s9**, **ciJy**). We retrain the open-source PaPaGei-S [1] model on our full pretraining corpus (“PaPaGei-Ours”) which trails MMR/MMR-Light on most tasks (Table 2), indicating that improvements derive from the proposed method rather than pretraining data alone.

*Chronos-Bolt*(Reviewer **z2s9**). We evaluated Chronos-Bolt [2], a recent time series foundation model, results and discussion in App A.5.2 and Table 3.

### **New Tasks**
*Demographic downstream tasks* (Reviewer **ciJy**). We added two demographic prediction tasks: age regression and age-group classification and report evaluation on our cohort for all baselines in Sec. 5; Tables 1–2; MMR performs strongly on these tasks.

*Public-dataset validation* (Reviewer **ciJy**). We incorporated additional public PPG downstream dataset DREAMT [4] where we added 4 sleep-stage classification tasks, and results are presented in Tables 1–2. MMR improves over multiple baselines, demonstrating generalization to a public, independently collected dataset.

### **Contribution Beyond Existing Works**
To address the concerns raised by (Reviewer **ciJy** and **H7f6**), we highlight contributions:

*Novel DWT Formulation.* We provide detailed discussion on why a wavelet-based transform is well-suited for modelling physiological biosignals such as PPG in App A.3. This novel formulation has not been explored in prior frequency-aware MAE variants.

*Comparison to PaPaGei [1].*  We find that MMR performs strongly compared to the pretrained PaPaGei and  PaPaGei model trained from scratch on our proprietary dataset (PaPaGei-Ours). We attribute these gains to the wavelet-based reconstruction task (see App A.3). Furthermore, we leverage a masked reconstruction objective that performs better than the contrastive learning approach for our downstream tasks. This is also evidenced by the performance gains of the time-domain masked reconstruction baseline (MTR) that we train on our dataset (Table 1) over PaPaGei-Ours.

*Comparison to Abbaspourazad et al [3].* This work does not present results of their method on public evaluation datasets. In MMR we present evaluation across a suite of 19 downstream tasks spread across 5 datasets (4 tasks from public dataset). We also provide comparison to multiple state-of-the-art foundation models (such as PaPaGei and Chronos).

### **Reproducibility/Additional Details.**
Reviewers **z2s9** **ciJy** and **kLnH** raised concerns about missing details. In the revised manuscript, we have added comprehensive descriptions of data preprocessing (App. A.6.1), the pretraining dataset and evaluation protocol (App. A.7 and A.6.4), and reproducibility and ethics statements (App. A.1 and A.2).

### **New Metrics/Ablations**
In response to Reviewer **H7f6**’s request, we have extended our analysis to provide deeper insight into the design choices of MMR. Specifically, we added (i) frequency-guided masking ablations (App Fig. 7a); (ii) design-choice stress tests (Fig. 6, App Sec. A.4), and (iii) additional evaluation metrics, including F1 scores for rare classes (Table 8,9 in App A.11).

### **Decomposition Depth Clarification**
Reviewer **kLnH** questioned whether task-specific optimal decomposition depth undermines representation transferability. We clarify that our main results use one fixed MMR setup (Level 3, Haar), pretrained once on the full dataset and applied unchanged to all 19 tasks, where it performs consistently well (Tables 1–2). The depth sweep was only part of a small-model, small-data ablation which is clarified in Sec. 5.4.

---

> ### Author Response · Authors · 2025-12-03
> **Summary of Reviews/Responses for New Area Chair**
>
> [1] PaPaGei: Open Foundation Models for Optical Physiological Signals, ICLR 2025
>
> [2] Chronos: Learning the Language of Time Series, TMLR 2024
>
> [3] Large-scale training of foundation models for wearable biosignals.ICLR 2023
>
> [4] Wang, Ke, et al. "DREAMT: Dataset for Real-time sleep stage Estimation using Multisensor wearable Technology." PhysioNet https://doi.org/10.13026/62AN-CB28 (2024).

---

### Meta-Review · Area_Chair_Ad5s · 2025-12-28

**Summary:**

The reviewers have raised the following major concerns:

(1) The core technical contribution lacks novelty, which makes the work incremental in this regard.

(2) Interpretation of model parameters (fine-grained waveform morphology v.s. global rhythmic dynamics) and the results.

(3) For a foundational model, different tasks should not benefit from different wavelet decomposition depths.

(4) Missing more recent baselines; missing discussion on performance gap.

(5) Missing details on model specifications.

(6) Data balance and diversity.

(7) Sampling and frequency balance.

**Reviewer Concerns:**

I believe the revision has addressed several major concerns raised by the reviewers (e.g., (3), (4), (5), and (6)). In particular, the numerical studies in the revised manuscript are more thorough and substantial, which improves the overall clarity and empirical support of the paper.

At the same time, I share the view of most reviewers that the technical contribution of the paper remains limited. The architectural design appears largely standard, and the manuscript does not provide sufficient theoretical or analytical discussion to clarify why the proposed framework should be expected to outperform existing approaches. While the authors note that “our backbone is intentionally standard ...” (see the rebuttal to Reviewer z2s9), it would be helpful to more clearly articulate other sources of novelty, for example, in data curation, model configuration, or evaluation methodology.

In addition, the discussion of model explanation and interpretation remains underdeveloped (see comments (2) by Reviewer kLnH and (7) by Reviewer H7f6). Given the positioning of the work, strengthening this aspect would be particularly important.

Finally, although the paper presents the model as foundational, the current scope of applications appears relatively narrow. The model is not trained across multiple modalities, and the diversity of the data types considered is limited. Expanding or better justifying the scope could further strengthen the paper.

**Reviewer Scores:**

There has been limited discussion during the rebuttal period, so it is uncertain whether the reviewers will revise their scores. Nonetheless, I believe there remain several concerns that could be addressed to further strengthen the paper.

---

### Decision · Program_Chairs · 2026-01-26

Reject